

# Air temperature equation derived from sonic temperature and water vapor mixing ratio for air flow sampled through closed-path eddy-covariance flux systems

Xinhua Zhou[1,2,3,4], Tian Gao[1,3,5*], Eugene S. Takle[3,7], Xiaojie Zhen[3,6], Andrew E. Suyker[4], Tala Awada[4], Jane Okalebo[4], Jiaojun Zhu[1,3,5]

[1]CAS Key Laboratory of Forest Ecology and Management, Institute of Applied Ecology, Chinese Academy of Sciences (CAS), Shenyang, 110016, China
[2]Campbell Scientific Inc., Logan, Utah, USA
[3]CAS-CSI Joint Laboratory of Research and Development for Monitoring Forest Fluxes of Trace Gases and Isotope Elements, Institute of Applied Ecology, CAS, Shenyang, China
[4]University of Nebraska, Lincoln, Nebraska, USA
[5]Qingyuan Forest CERN, CAS, Shenyang, China
[6]Beijing Techno Solutions Ltd., Beijing, China
[7]Iowa State University, Ames, Iowa, USA

*Correspondence to:* Tian Gao (tiangao@iae.ac.cn)

**Abstract.** Air temperar ($T$) plays a fundamental role in many aspects of the flux exchanges between the atmosphere and ecosystems. Additionally, it is critical to know where (in relation to other essential measurements) and at what frequency $T$ must be measured to accurately describe such exchanges. In closed-path eddy-covariance (CPEC) flux systems, $T$ can be computed from the sonic temperature ($T_s$) and water vapor mixing ratio that are measured by the fast-response senosrs of three-dimensional sonic anemometer and infrared gas analyzer, respectively. $T$ then is computed by use of either $T = T_s\left(1 + 0.51q\right)^{-1}$, where $q$ is specific humidity, or $T = T_s\left(1 + 0.32\,e/P\right)^{-1}$, where $e$ is water vapor pressure and $P$ is atmospheric pressure. Converting $q$ and $e/P$ into the same water vapor mixing ratio analytically reveals the difference between these two equations. This difference in a CPEC system could reach $\pm 0.18$ K, bringing an uncertainty into the accuracy of $T$ from both equations and raises the question of which equation is better. To clarify the uncertainty and to answer this question, the derivation of $T$ equations in terms of $T_s$ and $H_2O$-related variables is thoroughly studied. The two equations above were developed with approximations. Therefore, neither of their accuracies were evaluated, nor was the question answered. Based on the first principles, this study derives the $T$ equation in terms of $T_s$ and water vapor molar mixing ratio $\left(\chi_{H_2O}\right)$ without any assumption and approximation. Thus, this equation itself does not have any error and the accuracy in $T$ from this equation (equation-computed $T$) depends solely on the measurement accuracies of $T_s$ and $\chi_{H_2O}$. Based on current specifications for $T_s$ and $\chi_{H_2O}$ in the CPEC300 series and given their maximized measurement uncertainties, the accuracy in equation-computed $T$ is specified within $\pm 1.01$ K. This accuracy uncertainty is propagated mainly ($\pm 1.00$K) from the uncertainty in $T_s$ measurements and little ($\pm 0.03$K) from the uncertainty in $\chi_{H_2O}$ measurements. Apparently, the improvement on measurement technologies particularly for $T_s$ would be a key to narrow this accuracy range. Under normal sensor and weather conditions, the specified accuracy is overestimated and actual accuracy is better. Equation-computed $T$ has frequency response equivalent to high-frequency $T_s$ and is insensitive to solar contamination during measurements. As synchronized at a temporal scale of measurement frequency and matched at a spatial scale of measurement volume with all aerodynamic and thermodynamic variables, this $T$ has its advanced merits in boundary-layer meteorology and applied meteorology.



**Keywords:** $CO_2$ flux, $H_2O$ flux, infrared gas analyzer, sonic anemometer, turbulence measurements.

**1 Introduction**


The equation of state, $P = \rho RT$, is a fundamental equation for describing all atmospheric flows where $P$ is atmospheric pressure, $\rho$ is moist air density, $R$ is gas constant for moist air, and $T$ is air temperature (Wallace and Hobbs, 2006). In the boundary-layer, where turbulence is nearly always present, accurate representation of the "state" of the atmosphere at any given "point" and time requires consistent representation of spatial and temporal scales for all thermodynamic factors of $P$, $\rho$, and $T$ (Panofsky and


Dotton, 1984). Additionally, for observing fluxes describing exchanges of quantities such as heat and moisture between the earth and the atmosphere, it is critical to know all three-dimensional (3-D) components of wind speed at the same location and temporal scale as the thermodynamic variables (Laubach and McNaughton, 1998).

In a closed-path eddy-covariance (CPEC) system, the 3-D wind components and sonic temperature ($T_s$) are measured by a 3-D sonic anemometer in the sonic measurement volume near which air is sampled through the orifice of an infrared gas analyzer


into its closed-path $H_2O/CO_2$ measurement cuvette where air moisture is measured by the analyzer (Fig. 1). The flow pressure inside cuvette ($P_c$) and the differential ($\Delta P$) between $P_c$ and ambient flow pressure in the sampling location are also measured (Campbell Scientific Inc., 2018c). Atmospheric $P$ in the sampling volume, therefore, is a sum of $P_c$ and $\Delta P$. $P_c$, along with the internal $T$, are further used for infrared measurements of air moisture (i.e. $\rho_w$, $H_2O$ density) to calculate the water mixing ratio ($\chi_w$) inside the cuvette that is also equal to $\chi_w$ in the CPEC measurement volume including sonic measurement volume and air


sampling location. Finally, the $T_s$ and $\chi_w$ from the CPEC measurement volume, after spatial and temporal synchronization (Horst and Lenschow, 2009), are used to calculate the $T$ inside this volume. Two optional equations (Schotanus et al., 1983; Kaimal and Gaynor, 1991; see the section of Background), which need rigorous evaluation, are available for this $T$ calculation. In summary, the boundary-layer flow measured by a CPEC system has all variables quantified with consistent representation of spatial and temporal scales for moist turbulence thermodynamics (i.e. state) if the following are available: the 3-D wind; $P$ measured


differentially; $T$ from an equation; and $\rho$ from $P$, $T$, and $\chi_w$.





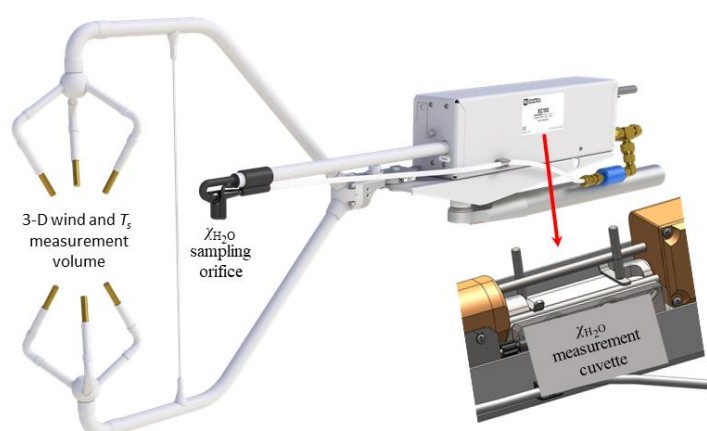

**Figure 1: Measurement volume for three-dimensional (3-D) wind and sonic temperature ($T_s$), sampling orifice for H₂O mixing ratio ($\chi_{H_2O}$), and measurement cuvette for $\chi_{H_2O}$ in CPEC300 series (Campbell Scientific Inc., UT, USA).**

In this paper, we 1) derive a $T$ equation in terms of $T_s$ and $\chi_w$ based on the first principles as an alternative to the commonly

used equations which are based on approximations; 2) estimate and verify the accuracy of the first-principles $T$; 3) assess the expected advantages of the first-principles $T$ as a high frequency signal insensitive to solar contamination suffered to conventional $T$ sensor measurements (Lin et al., 2001; Blonquist and Bugbee, 2018), and 4) address the potential applications of the derived $T$ equation in flux measurements. We first provide a brief summary of the moist turbulence thermodynamics of the boundary-layer flows measured by CPEC flux systems.

**2 Background**

A CPEC system commonly is used to measure boundary-layer flows for the $CO_2$, $H_2O$, heat, and momentum fluxes between ecosystems and the atmosphere. Such a system is equipped with a 3-D sonic anemometer to measure the speed of sound in three dimensions in the central open-space of the instrument (hereinafter, refer to open-space), from which can be calculated $T_s$ and 3-D components of wind at fast response. Integrated with this sonic anemometer, a fast-response infrared gas analyzer concurrently

measures $CO_2$ and $H_2O$ in its cuvette (closed-space) of infrared measurements, through which air is sampled under pump pressure while being heated (Fig. 1). The analyzer outputs $CO_2$ mixing ratio (i.e. $\chi_{CO_2} = \rho_{CO_2} / \rho_d$ where $\rho_{CO_2}$ is $CO_2$ density and $\rho_d$ is dry air density) and $\chi_w$ (i.e. $\rho_w / \rho_d$). Together these instruments provide high-frequency (e.g., 10 Hz) measurements from which the fluxes are computed (Aubinet et al., 2012) at a "point" represented by the sampling space of the CPEC system.

These basic high-frequency measurements of 3-D wind speed, $T_s$, $\chi_w$, and $\chi_{CO_2}$ provide observations from which mean and

fluctuation properties of air, such as $\rho_d$, $\rho$, $\rho_w$, $\rho_{co_2}$, and hence fluxes can be determined. For instance, water vapor flux is



calculated from $\overline{\rho_d w' \chi_w'}$ where $w$ is vertical velocity of air and prime indicates the fluctuation of variable away from its mean as indicated by overbar, e.g. $w' = w - \overline{w}$). Given the measurements of $\chi_w$ and $P$ from CPEC systems, based on the gas laws (Wallace and Hobbs 2006), $\rho_d$ is drived from

$$\rho_d = \frac{P}{T(R_d + R_v \chi_w)} \tag{1}$$

where $R_d$ is gas constant for dry air and $R_v$ is gas constant for water vapor. In turn, $\rho_w$ is equal to $\rho_d \chi_w$ and $\rho$ is a sum of $\rho_d$ and $\rho_w$. All mentioned physical properties can be derived if $T$ in Eq. (1) for $\rho_d$ is acquired.

Additionally, equations for ecosystem exchange and flux require $\overline{\rho_d}$ (Gu et al., 2012) and $\overline{\rho_d w}$ (Foken et al., 2012). Furthermore, due to accuracy limitations in measurements of $w$ from a modern sonic anemometer, the dry air flux of $\overline{\rho_d w}$ must be derived from $\overline{\rho_d' w'} - \overline{\rho_d}\,\overline{w}$ (Webb et al., 1980; Lee and Massman, 2011). Because of its role in flux measurements, a high frequency representation of $\rho_d$ is needed. To acquire such $\rho_d$ from Eq. 1 for advanced applications, high-frequency $T$ in temporal synchronization with $\chi_w$ and $P$ is needed.

In a modern CPEC system, $P$ is measured using a fast-response barometer suitable for measurements at a high frequency (e.g. 10 Hz, Campbell Scientific Inc., 2018a) and, as discussed above, $\chi_w$ is a high frequency signal from a fast-response gas analyzer (e.g. commonly up to 20 Hz). If $T$ is measured using a slow-response sensor, the three independent variables in Eq. (1) do not have equivalent synchronicity in frequency response. In terms of frequency response, $\rho_d'$ cannot be correctly aquired. $\overline{\rho_d}$ derived based on Eq. (1) also has uncertainty, although it can be approximated from either of two following equations:

$$\overline{\rho_d} = \overline{\frac{P}{T(R_d + R_v \chi_w)}} \tag{2}$$

and

$$\overline{\rho_d} = \frac{\overline{P}}{\overline{T}(R_d + R_v \overline{\chi_w})}. \tag{3}$$

Eq. (2) is mathematically valid in averaging rules (Stull, 1988), but the response of the system to $T$ is slower than to $\chi_w$, and even $P$ and the Eq. (3) is invalid under averaging rules although its three over-bar independent variables can be evaluated over an average interval. Consequently, neither $\overline{\rho_d w}$ nor $\overline{\rho_d}$ can be evaluated strictly in theory.

Measurements of $T$ at high frequency (similar to those at low-frequency) are contaminated by solar radiation, even under shields (Lin et al., 2001) and when aspirated (Campbell Scientific Inc., 2010; R.M. Young Company, 2004; Apogee Instrument Inc., 2013; Blonquist and Bugbee, 2018). Additionally, fine wires and aspiration methods have limited applicability for long-term measurements in rugged field conditions typically encountered in ecosystem monitoring.

To avoid the issues above in use of either slow- or fast-response $T$ sensors under field conditions, deriving $T$ from $T_s$ and $\chi_w$ (Schotanus et al., 1983; Kaimal and Gaynor, 1991) is an advantageous alternative to the applications of $T$ in CPEC measurements. In a CPEC system, $T_s$ is measured at a high frequency (e.g. 10 Hz) using a fast-response sonic anemometer to detect the speed of sound in the open-space (Munger et al., 2012) provided there is no evidence of contaminated by solar radiation. It is a high-frequency signal. $\chi_w$ is measured at the same frequency as for $T_s$ using a gas analyzer equivalent to the



sonic anemometer in high-frequency response time (Ma et al., 2017). $\chi_w$ reported from a CPEC system is converted from water vapor molar density measured inside the closed-cuvette whose internal pressure and internal temperature are more stable than $P$ and $T$ in the open-space and can be more accurately measured. Because of this, solar warming and radiation cooling of the

cuvette is irrelevant as long as water molar density, pressure, and temperature inside the closed-cuvette are more accurately measured. Therefore, it could be reasonably expected that $T$ calculated from $T_s$ and $\chi_w$ in a CPEC system should be a high-frequency signal insensitive to solar radiation.

The two equations commonly used to compute $T$ from $T_s$ and air moisture-related variables are, given by Schotanus et al. (1983),

$$T = T_s \left(1 + 0.51 q\right)^{-1}, \tag{4}$$

where $q$ is specific humidity, defined as a ratio of water vapor to moist air density, and by Kaimal and Gaynor (1991),

$$T = T_s \left(1 + 0.32 \frac{e}{P}\right)^{-1}, \tag{5}$$

where $e$ is water vapor pressure. Re-arranging these two equations gives $T$ in terms of $T_s$ and $\chi_w$. Expressing $q$ in terms of $\rho_d$ and $\rho_w$, Eq. (4) becomes

$$T = T_s \left(1 + 0.51 \frac{\rho_w}{\rho_d + \rho_w}\right)^{-1} = T_s \left(1 + 0.51 \frac{\chi_w}{1 + \chi_w}\right)^{-1}, \tag{6}$$

and expressing $e$ and $P$ using the equation of state, Eq. (5) becomes

$$T = T_s \left(1 + 0.32 \frac{R_v T \rho_w}{R_d T \rho_d + R_v T \rho_w}\right)^{-1} = T_s \left(1 + 0.51 \frac{\chi_w}{1 + 1.61 \chi_w}\right)^{-1}. \tag{7}$$

The $\chi_w$-related terms in the denominator inside parentheses in both equations above clearly reveal that $T$ values from the same $T_s$ and $\chi_w$ using the two commonly used Eqs. (4) and (5) will not be the same. The absolute difference in the values ($\Delta T_e$, i.e.

the difference in $T$ between Eqs. (4) and (5) can be analytically expressed as

$$\Delta T_e = \frac{0.31 T_s \chi_w^2}{1 + 3.63 \chi_w + 3.20 \chi_w^2}. \tag{8}$$

Given that, in a CPEC system, the sonic anemometer has an operational range in $T_s$ of -30 to 57 °C (Campbell Scientific Inc., 2018b) and a gas analyzer has a measurement range in $\chi_w$ of 0 to 0.045 kgH$_2$O kg$^{-1}$ (Campbell Scientific Inc., 2018a), $\Delta T_e$ ranges up to 0.177 K, which brings an uncertainty in accuracy of $T$ calculated from either Eq. (4) or (5) and raises the

question of which equation is better.

Reviewing the sources of Eq. (4) (Schotanus et al., 1983; Swiatek, 2009; van Dijk, 2002) and Eq. (5) (Ishii, 1932; Barrett and Suomi, 1949; Kaimal and Businger, 1963; Kaimal and Gaynor, 1991), it was found that approximation procedures were used in derivation of both equations, but the approach to the derivation of Eq. (4) (Appendix A) is different from that of Eq. (5) (Appendix B). The different approaches create a difference between the two commonly used equations as shown in Eq. (8), and

the approximation procedures lead to the controversy as to which equation is more accurate. The controversy can be avoided if the $T$ equation in terms of $T_s$ and $\chi_w$ can be derived from the $T_s$ equation and the first principles equations, if possible, without an approximation and verified against precision measurements of $T$ with minimized solar contamination.





### 3 Derivation of air temperature equation

As discussed above, a sonic anemometer measures the speed of sound ($c$) concurrent with measurement of the 3-D wind speed
(Munger et al., 2012). The speed of sound in the homogeneous atmospheric boundary-layer is defined by Barrett and Suomi
(1949):

$$c^2 = \gamma \frac{P}{\rho}, \tag{9}$$

where $\gamma$ is the ratio of moist air specific heat at constant pressure ($C_p$) to moist air specific heat at constant volume ($C_v$).
Substitution of the equation of state into Eq. (9), gives $T$ as a function of $c$:

$$T = \frac{c^2}{\gamma R}. \tag{10}$$

This equation reveals the opportunity to use a measured $c$ for the $T$ calculation; however, both $\gamma$ and $R$ depend on air humidity,
which is unmeasurable by a sonic anemometry itself; Eq. (10) is, therefore, not applicable for $T$ calculations inside a sonic
anemometer. Alternatively, $\gamma$ is replaced with its counterpart for dry air [$\gamma_d$, 1.4003, i.e. the ratio of dry air specific heat at
constant pressure ($C_{pd}$, 1,004 J K$^{-1}$ kg$^{-1}$) to dry air specific heat at constant volume ($C_{vd}$, 717 J K$^{-1}$ kg$^{-1}$)] and $R$ is replaced with its
counterpart for dry air ($R_d$, 287.06 J K$^{-1}$ kg$^{-1}$, i.e. gas constant for dry air). Both replacements make the right side of Eq. (10)
become $c^2/\gamma_d R_d$ which is no longer a measure of $T$. However, $\gamma_d$ and $R_d$ are close to their respective values of $\gamma$ and $R$ in
magnitude and the right side of Eq. (10) after the replacements is defined as sonic temperature ($T_s$), given by (Campbell
Scientific Inc., 2018b):

$$T_s = \frac{c^2}{\gamma_d R_d}. \tag{11}$$

Comparing this equation to Eq. (10), given $c$, if air is dry, $T$ must be equal to $T_s$; therefore, we define "*sonic temperature of moist
air is the temperature that its dry air component reaches when moist air has the same enthalpy*". Since both $\gamma_d$ and $R_d$ are
constants and $c$ is measured by a sonic anemometer and corrected for crosswind effect inside the sonic anemometer based on its
3-D wind measurements (Liu et al., 2001; Zhou et al., 2018), Eq. (11) is used inside the operating system of modern sonic
anemometers to report $T_s$ instead of $T$.

Equations (9) to (11) provide a theoretical basis of first principles to derive the relationship of $T$ to $T_s$ and $\chi_w$. In Eq. (9), $\gamma$ and
$\rho$ vary with air humidity and $P$ is related to $\rho$ as described by the equation of state. Consequently, the derivation of $T$ from $T_s$
and $\chi_w$ for a CPEC system needs to address the relationship of $\gamma$, $\rho$, and $P$ to air humidity in terms of $\chi_w$.

### 3.1 Relationship of $\gamma$ to $\chi_w$

For moist air, the ratio of specific heat at constant pressure to specific heat at constant volume is:

$$\gamma = \frac{C_p}{C_v}, \tag{12}$$

where $C_p$ varies with air moisture between $C_{pd}$ and $C_{pw}$ (water vapor specific heat at constant pressure, 1,952 J kg$^{-1}$ K$^{-1}$). It is the
arithmetical average of $C_{pd}$ and $C_{pw}$ weighted by dry air mass and water vapor mass, respectively, given by (Stull, 1988; Swiatek,
2009):



$$C_p = \frac{C_{pd}\rho_d + C_{pw}\rho_w}{\rho_d + \rho_w} . \tag{13}$$

Based on the same rationale, $C_v$ is:

$$C_v = \frac{C_{vd}\rho_d + C_{vw}\rho_w}{\rho_d + \rho_w} , \tag{14}$$

where $C_{vw}$ is the specific heat of water vapor at constant volume (1,463 J kg$^{-1}$ K$^{-1}$). Substituting Eqs. (13) and (14) into Eq. (12) generates:

$$\gamma = \gamma_d \frac{1 + \left(C_{pw}/C_{pd}\right)\chi_w}{1 + \left(C_{vw}/C_{vd}\right)\chi_w} \tag{15}$$

**3.2 Relationship of $P/\rho$ to $\chi_w$**

Atmospheric $P$ is the sum of $P_d$ and $e$. Similarly, $\rho$ is the sum of $\rho_d$ and $\rho_w$. Using the equation of state, the ratio of $P$ to $\rho$ can be expressed as:

$$\frac{P}{\rho} = \frac{R_d T \rho_d + R_v T \rho_w}{\rho_d + \rho_w} = \frac{R_d T\left(1 + \dfrac{R_v}{R_d}\chi_w\right)}{1 + \chi_w} . \tag{16}$$

In this equation, the ratio of $R_v$ to $R_d$ is given by:

$$\frac{R_v}{R_d} = \frac{R^*/M_w}{R^*/M_d} = \frac{1}{M_w/M_d} , \tag{17}$$

where $R^*$ is the universal gas constant, $M_w$ is the molecular mass of water vapor (18.0153 kg kmol$^{-1}$), $M_d$ is the molecular mass of dry air (28.9645 kg kmol$^{-1}$). The ratio of $M_w$ to $M_d$ is 0.622, conventionally denoted by $\varepsilon$. Substituting Eq. (17), after its denominator is represented by $\varepsilon$, into Eq. (16) leads to:

$$\frac{P}{\rho} = \frac{R_d T\left(\varepsilon + \chi_w\right)}{\varepsilon\left(1 + \chi_w\right)} . \tag{18}$$

**3.3 Relationship of $T_s$ to $T$ and $\chi_w$**

Substituting Eqs. (15) and (18) into Eq. (9), $c^2$ is expressed in terms of $T$ and $\chi_w$ along with atmospheric physics constants. Further, substituting $c^2$ into Eq. (11) generates:

$$c^2 = \frac{R_d \gamma_d T\left(\varepsilon + \chi_w\right)\left[1 + \left(C_{pw}/C_{pd}\right)\chi_w\right]}{\varepsilon\left(1 + \chi_w\right)\left[1 + \left(C_{vw}/C_{vd}\right)\chi_w\right]} \tag{19}$$

Further, substituting $c^2$ into Eq. (11) generates:

$$T_s = T\frac{\left(\varepsilon + \chi_w\right)\left[1 + \left(C_{pw}/C_{pd}\right)\chi_w\right]}{\varepsilon\left(1 + \chi_w\right)\left[1 + \left(C_{vw}/C_{vd}\right)\chi_w\right]} . \tag{20}$$

Now, this equation expresses $T_s$ in terms of $T$ of interest to this study, $\chi_w$, and atmospheric physics constants (i.e. $\varepsilon$, $C_{pw}$, $C_{pd}$,





$C_{vw}$, and $C_{vd}$).

### 3.4 Air temperature equation

Rearranging the terms in Eq. (20) results in

$$T = T_s \frac{\varepsilon(1+\chi_w)\left[1+\left(C_{vw}/C_{vd}\right)\chi_w\right]}{(\varepsilon+\chi_w)\left[1+\left(C_{pw}/C_{pd}\right)\chi_w\right]}$$
(21)

This equation shows that $T$ is a function of $T_s$ and $\chi_w$ that are measured at high frequency in a CPEC system by a sonic anemometer and a gas analyzer.

A CPEC system outputs water vapor molar mixing ratio (Campbell Scientific Inc., 2018a) commonly used in the community of eddy-covariance fluxes (AmeriFlux, 2018). The relation of water vapor mass to molar mixing ratio ( $\chi_{H_2O}$ in molH$_2$O mol$^{-1}$)

is given by:

$$\chi_w = \frac{M_w}{M_d}\chi_{H_2O} = \varepsilon\chi_{H_2O}.$$
(22)

Substituting this relation into Eq. (21) and denoting $C_{vw}/C_{vd}$ with $\gamma_v = 2.04045$ and $C_{pw}/C_{pd}$ with $\gamma_p = 1.94422$, Eq. (21) is expressed as:

$$T = T_s \frac{\left(1+\varepsilon\chi_{H_2O}\right)\left(1+\varepsilon\gamma_v\chi_{H_2O}\right)}{\left(1+\chi_{H_2O}\right)\left(1+\varepsilon\gamma_p\chi_{H_2O}\right)}.$$
(23)

This is the air temperature equation in terms of $T_s$ and $\chi_{H_2O}$ for use in CPEC systems. It is derived from a theoretical basis of first principles [i.e. Eqs. (9) to (11)]. In its derivation, except for the use of the equation of state and Dalton's law, no other assumptions nor approximations are used. Therefore, Eq. (23) is an exact equation of $T$ in terms of $T_s$ and $\chi_{H_2O}$ for the air flow sampled through a CPEC system and avoids the controversy in use of Eqs. (4) and (5) arising from approximations as shown in Appendices A and B. Therefore, $T$ computed from this equation (hereinafter referred to as equation-computed $T$) should be

accurate as long as the values of $T_s$ and $\chi_{H_2O}$ are exact.

For this study, however, $T_s$ and $\chi_{H_2O}$ are measured by the CPEC systems deployed in the field under changing weather conditions. Their measured values must include measurement uncertainty in $T_s$, denoted by $\Delta T_s$ and in $\chi_{H_2O}$ as well, denoted by $\Delta\chi_{H_2O}$. The uncertainties, $\Delta T_s$ and/or $\Delta\chi_{H_2O}$, unavoidably propagate to create uncertainty in equation-computed $T$, denoted by $\Delta T$, which makes an exact $T$ impossible. In numerical analysis (Burden and Faires, 1993) or in statistics (Snedecor and

Cochran, 1991), any applicable equation requires the specification of an uncertainty term. Therefore, the equations for $T$ should include specification of their respective uncertainty expressed as the bounds (i.e. the maximum and minimum limits) specifying the range of the equation-computed $T$ that need to be known for any application. According to the definition of accuracy that was advanced by the International Organization for Standardization (2012), this uncertainty range is equivalent to the "accuracy" as the range of both trueness and precision (i.e. total uncertainty). This accuracy ($\Delta T$) should be specified through its relationship to

$\Delta T_s$ and $\Delta\chi_{H_2O}$.





### 3.5 Relationship of $\Delta T$ to $\Delta T_s$ and $\Delta \chi_{H_2O}$

$\Delta T_s$ and $\Delta \chi_{H_2O}$ are the measurement accuracies which can be reasonably considered as small increments in a calculus sense. As such, depending on both small increments, $\Delta T$ is the total differential of $T$ with respect to $T_s$ and $\chi_{H_2O}$, given by:

$$\Delta T = \frac{\partial T}{\partial T_s} \Delta T_s + \frac{\partial T}{\partial \chi_{H_2O}} \Delta \chi_{H_2O} \tag{24}$$

The two partial derivatives in the right side of this equation can be derived from Eq. (23). Substituting the two partial derivatives into this equation leads to

$$\Delta T = \frac{T}{T_s} \Delta T_s + T \left[ \frac{\varepsilon + \varepsilon \gamma_v \left( 1 + 2\varepsilon \chi_{H_2O} \right)}{\left( 1 + \varepsilon \chi_{H_2O} \right)\left( 1 + \varepsilon \gamma_v \chi_{H_2O} \right)} - \frac{1 + \varepsilon \gamma_p \left( 1 + 2\chi_{H_2O} \right)}{\left( 1 + \chi_{H_2O} \right)\left( 1 + \varepsilon \gamma_p \chi_{H_2O} \right)} \right] \Delta \chi_{H_2O} \tag{25}$$

This equation indicates that in dry air when $T = T_s$, $\Delta T$ is equal to $\Delta T_s$ if $\chi_{H_2O}$ is measured accurately (i.e. $\Delta \chi_{H_2O} = 0$ while $\chi_{H_2O} = 0$). However, air in the atmospheric boundary-layer where CPEC systems are used is always moist. Given this

equation, $\Delta T$ at $T_s$ and $\chi_{H_2O}$ can be evaluated by using $\Delta T_s$ and $\Delta \chi_{H_2O}$, both of which are related to the measurement specifications of sonic anemometer for $T_s$ (Campbell Scientific Inc., 2018b) and of gas analyzer for $\chi_{H_2O}$ (Campbell Scientific Inc., 2018a). In the right side of Eq. (25), the first term with $\Delta T_s$ can be expressed as $\Delta T_{T_s}$ (uncertainty portion of $\Delta T$ due to $\Delta T_s$) and the second with $\Delta \chi_{H_2O}$ can be expressed as $\Delta T_{\chi_{H_2O}}$ (uncertainty portion of $\Delta T$ due to $\Delta \chi_{H_2O}$). Using $\Delta T_{T_s}$ and $\Delta T_{\chi_{H_2O}}$, this equation can be simplified as:

$$\Delta T = \Delta T_{T_s} + \Delta T_{\chi_{H_2O}} \tag{26}$$

The range of $\Delta T$ from this equation essentially is the accuracy of equation-computed $T$.

### 4 Accuracy of equation-computed $T$

The CPEC system for this study was CPEC310 (Campbell Scientific Inc., Logan, UT, USA) including a CSAT3A sonic anemometer (updated version in 2016) for fast response to 3-D wind and $T_s$ and an EC155 gas analyzer for fast response to $H_2O$

along with $CO_2$ (Burgon et al., 2015; Ma et al., 2017). The system operates in a $T$ range of -30 to 50 °C and measures $\chi_{H_2O}$ in a range up to 79 mmol mol[-1] (i.e. 37 °C dew point temperature at 86 kPa under manufacturer environment); therefore, the accuracy of equation-computed $T$, depending on $\Delta T_s$ and $\Delta \chi_{H_2O}$, should be defined and estimated in a domain over both ranges.

### 4.1 $\Delta T_s$ (measurement accuracy in $T_s$)

As is true for other sonic anemometers (e.g. Gill Instruments, 2004), the CSAT3A has not been assigned a $T_s$ measurement
performance (Campbell Scientific Inc., 2018b) because the theories and methodologies how to specify this performance, to the best of our knowledge, have not been clearly defined. The performance of CSAT series for $T_s$ is best near production temperature around 20 °C and drifts little away from this temperature. Within the operational range of temperature, the updated version of



CSAT3A has an overall uncertainty to be ± 1.0 °C (i.e. $\left|\Delta T_s\right| < 1.0 \text{ K}$, personal communication with CSAT authority: Larry Jacobsen through email in 2017 and in his office in 2018).

**4.2 $\Delta\chi_{H_2O}$ (measurement accuracy in $\chi_{H_2O}$)**

$\Delta\chi_{H_2O}$ is the accuracy in $H_2O$ measurements from gas analyzers, depending on analyzer measurement performance. This performace is specified using four component uncertainties: precision $\left(\sigma_{H_2O}\right)$, maximum zero drift with ambient air temperature $(d_{wz})$, maximum gain drift with ambient air temperature $(d_{wg})$, and cross-sensitivity to $CO_2$ $(s_{wc})$ (LI-COR Bioscience, 2016; Campbell Scientific Inc., 2018c). Adopting the newly advanced definition of accuracy as the full range of overall uncertainty in measurements (International Organization for Standardization, 2012), Li et al. (2021) composited the four component uncertainties as an accuracy of $H_2O$ field measurements in ecosystems from gas analyzers, given by:

$$\Delta\chi_{H_2O} = \pm\left[1.96\sigma_{H_2O} + 600s_{wc} + \left(d_{wz} + d_{H_2O\_gp}\chi_{H_2O}\right)\times\begin{cases}\dfrac{T_c - T_{zs}}{T_{rh} - T_{zs}} & T_{zc} < T_c \leq T_{rh} \\[2ex] \dfrac{T_{zs} - T_c}{T_{zc} - T_{rl}} & T_{zc} > T_c \geq T_{rl}\end{cases}\right], \tag{27}$$

where $d_{H2O\_gp}$ is gain drit percent (0.3%) and $T_{zs}$ is $T_c$ at which a gas analyzer was calibrated by the manufacturer to fit its working equation or zeroed/spanned in the field to adjust the zero/gain drift, and $T_{rl}$ and $T_{rh}$ are the low- and the high-end values, respectively, over the operational air temperature range of CPEC systems. Given the gas analyzer specifications: $\sigma_{H_2O}$, $d_{wz}$, $d_{H2O\_gp}$, $s_{wc}$, $T_{rh}$, and $T_{rl}$; this equation can be used to estimate $\Delta\chi_{H_2O}$ in Eq. (25) eventually for $\Delta T_{\chi_{H_2O}}$ in Eq. (26) over the domain of $T$ and $\chi_{H_2O}$.

**4.3 $\Delta T$ (Accuracy in equation-computed $T$)**

$\Delta T$ can be evaluated using $\Delta T_s$ and $\Delta\chi_{H_2O}$ (Eq. 25) varying with $T$, $T_s$, and $\chi_{H_2O}$. Both $T$ and $T_s$ reflect air temperature, being associated each other through $\chi_{H_2O}$ (Eq. 23). Given $\chi_{H_2O}$, $T$ can be calculated from $T_s$, and vice versa; therefore, for the figure presentations in this study, it is sufficient to use either $T$ or $T_s$, instead of both, to show $\Delta T$ with air temperature. Considering $T$ to be of interest to this study, $T$ should be used. As such, $\Delta T$ can be analyzed over a domain of $T$ and $\chi_{H_2O}$ within the CPEC operational range of $T_c$ from -30 to 50 °C across the analyzer measurement range of $\chi_{H_2O}$ from 0 to 0.079 $molH_2O$ $mol^{-1}$.

To visualize the relationship of $\Delta T$ with $T_c$ and $\chi_{H_2O}$, $\Delta T$ is presented better as ordinate along $T_c$ as abscissa associated with $\chi_{H_2O}$. However, due to the positive dependence of air water vapor saturation on $T$ (Wallace and Hobbs, 2006), $\chi_{H_2O}$ has a possible range wider at higher $T_c$ and narrower at lower $T_c$. To present $\Delta T$ over the same measure of air moisture even at different $T$, the saturation water vapor pressure is used to scale air moisture to 0, 20, 40, 60, 80 and 100 (i.e. RH, relative humidity in %). For each scaled RH value, $\chi_{H_2O}$ can be calculated at different $T_c$ and $P$ (Appendix C) for use of Eq. (25). In this way, over the range of $T_c$, the trend of $\Delta T$ due to each measurement uncertainty source can be shown along the curves with equal





RH as the measure of air moisture (Fig. 2).

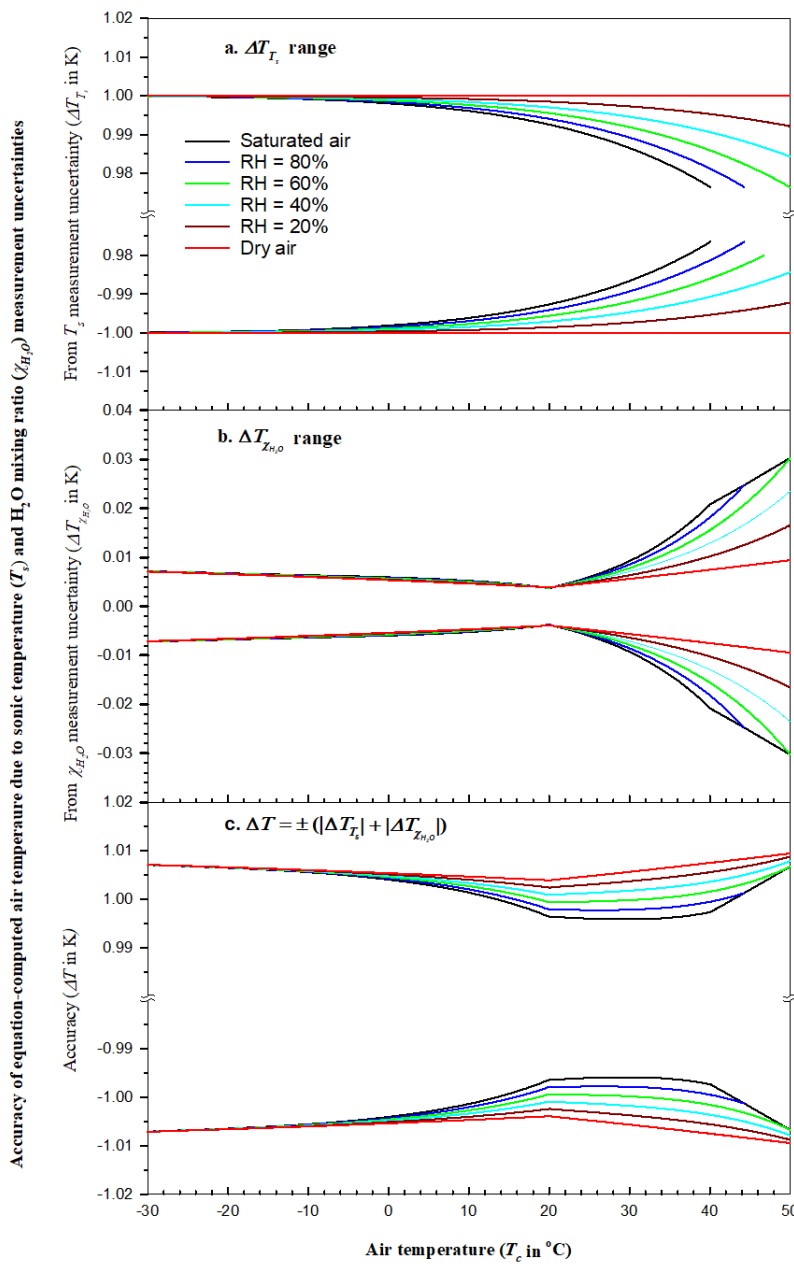

Figure 2: Accuracy of air temperature computed from Eq. (23) (equation-computed $T$) over the measurement range of $H_2O$ mixing ratio within the operational range of air temperature for CPEC300 series (Campbell Scientific Inc., UT, USA). a. Accuracy component of equation-computed $T$ due to sonic temperature measurement uncertainty, b. Accuracy component of equation-computed $T$ due to $H_2O$ mixing ratio uncertainty, and c. the overall accuracy of equation-computed $T$.






### 4.3.1 $\Delta T_{T_s}$ (Uncertainty portion of $\Delta T$ due to $\Delta T_s$ )

Given $\Delta T_s = \pm 1.00$ K and $T_s$ from the algorithm in Appendix C, $\Delta T_{T_s}$ in Eq. (26) was calculated over the domain of $T_c$

and $\chi_{H_2O}$ (Fig. 2a). Over the whole $T$ range, the $\Delta T_{T_s}$ limits range $\pm 1.00$ K, becoming a little narrower with $\chi_{H_2O}$ increasing due

to decrease, at the same $T_s$, in the magnitude $T/T_s$ in Eq. (25). The narrowest limits of $\Delta T_{T_s}$, in an absolute value, varies $< 0.01$

K over the range of $T$ below 20 °C although $> 0.01$ K above, but $< 0.03$ K.

### 4.3.2 $\Delta T_{\chi_{H_2O}}$ (Uncertainty portion of $\Delta T$ due to $\Delta\chi_{H_2O}$ )

Given $\Delta\chi_{H_2O}$ from Eq. (27) and $\chi_{H_2O}$ from the algorithm in Appendix C, $\Delta T_{\chi_{H2O}}$ was calculated over the domain of $T_c$

and $\chi_{H_2O}$ (Fig. 2b). The parameters in Eq. (27) are given through the specifications of EC155 (Campbell Scientific Inc., 2018c,

$\sigma_{H_2O}$ is $6.0\times10^{-6}$ molH2O mol$^{-1}$ where mol is a unit for dry air; $d_{wz}$, $\pm5.0\times10^{-5}$ molH2O mol$^{-1}$ with $T_c$, $d_{cg}$, $\pm0.30\%$ $\chi_{H_2O}$ in

molH2O mol$^{-1}$ with with $T_c$; $s_{wc}$, $\pm5.0\times10^{-8}$ molH2O mol$^{-1}$ ($\mu$molCO2 mol$^{-1}$)$^{-1}$ $T_{zc}$, 20 °C; $T_{rl}$, -30 °C; and $T_{rh}$, 50 °C).

$\Delta T_{\chi_{H2O}}$ tends to be smallest at $T_c = T_{zc}$. However, away from $T_{zc}$, its range non-linearly becomes wider, very gradually below

$T_{zc}$ but more abruptly above, because, as temperature increases, $\chi_{H_2O}$ at the same RH increases exponentially (Eqs. c1 and c5)

while $\Delta\chi_{H_2O}$ increases linearly with $\chi_{H_2O}$ in Eq. (27). This non-linear range can be summarized to be $\pm0.01$ K below 30 °C

and $\pm0.03$ K above 30 °C. Compared to $\Delta T_{T_s}$, $\Delta T_{\chi_{H2O}}$ is much smaller at two orders. $\Delta T_{T_s}$ is a large compoonent in $\Delta T$ .

### 4.3.3 $\Delta T$ (Combined uncertainty as the accuracy in equation-computed $T$)

Equation (26) is used to determine the maximum combined uncertainty in equation-computed $T$ for the same RH grade in Fig. 2

by adding together the same sign curve data of $\Delta T_{T_s}$ in panel a and $\Delta T_{\chi_{H2O}}$ in panel b. $\Delta T$ ranges at different RH grades are

shown in panel c. This panel can specify the accuracy of equation-computed $T$ at 101.325 kPa (i.e. atmospheric pressure at the

sea level at 15 °C) over the $\chi_{H_2O}$ measurement range to be within $\pm1.01$ K.

### 4.4 Accuracy of equation-computed $T$ from CPEC field measurements

Equation (23) is derived particularly for CPEC systems in which $T_s$ and $\chi_{H_2O}$ are measured neither at the same volume nor at the

same time. Both variables are measured separately using a sonic anemometer and a gas analyzer in a spatial separation between

the $T_s$ measurement center and the $\chi_{H_2O}$ measurement cuvette (e.g. Fig. 1) along with a temporal lag in the measurement

of $\chi_{H_2O}$ relative to $T_s$ due to the transport time and phase shift (Ibrom et al., 2007) of air flows sampled for $\chi_{H_2O}$ through the

sampling orifice to the measurement cuvette (Fig. 3).



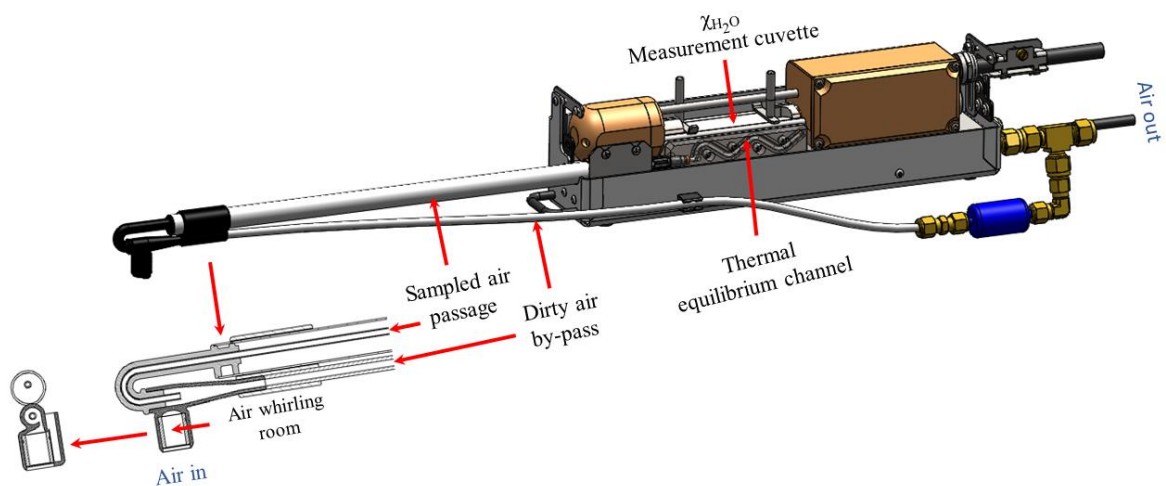

**Figure 3: Vortex intake system for air flow through its individual compartments: Air whirling room (2.200 mL), sampled air passage (1.889 mL), thermal equilibrium channel (0.587 mL), and $\chi_{H_2O}$ measurement cuvette (5.887 mL). The internal space of all**

**compartments adds up to a total volume of 10.563 mL.**

Fortunately, the spatial separation is at the tens of centimeter scale and the temporal lag is of the tens of millisecond scale. In eddy-covariance flux measurements, such a separation misses some covariance signals at higher frequency, which is correctable (Moore, 1986), and such a lag diminishes the covariance correlation, which is recoverable (Ibrom et al., 2007). How such a separation along with the lag influences the accuracy of Eq. (23), as shown in Fig. 2, needs testing against precision

measurements of air temperature. The two advantages of the equation-computed $T$ discussed in the introduction, namely the fast response to high frequency signals and the insensitivity to solar contamination in measurements, were studied and assessed during testing when a CPEC system was set up in the Campbell Scientific Instrument Test Field (41.8° N, 111.9° W, 1,360 m asl, UT, USA).

## 5 Field test station

A CPEC310 system was set as a core of the station in 2018. Beyond its major components briefly described in Section 4, the system also included a barometer (Model: MPXAZ6115A, Freescale Semiconductor, TX, USA) for flow pressure, pump module (SN: 1001) for air sampling, valve module (SN: 1003) to control flows for auto zero/span $CO_2$ and $H_2O$, scrub module (SN: 1002) to generate zero gas (i.e. without $CO_2$ and $H_2O$) for auto zero procedure, a $CO_2$ cylinder for $CO_2$ span, and an EC100 electronic module (SN: 1002, OS: Rev 07.01) to control and measure CSAT3A, EC155, and a barometer. In turn, the EC100 was connected

to, and instructed by, a central CR6 Datalogger (SN: 2981, OS 04) for these sensor measurements, data processing, and data output. In addition to receiving the data output from the EC100, the CR6 also controlled the pump, valve, and scrub modules and measured other micrometeorological sensors in support for this study.

The micrometeorological sensors included a LI200 pyranometer (SN: 18854, LI-Cor Biosciences, Lincoln, NE, US) to



monitor incoming solar radiation, a precision platinum resistance temperature detector (RTD, model 41342, SN: TS25360)

inside a fan-aspirated radiation shield (model: 43502, R.M. Young Company, Traverse City, MI, USA) to more accurately measure the $T$ considered with minimized solar contamination due to higher fan-aspiration efficiency, and a HMP155A temperature and humidity senor (SN: 1073, Vaisala Corporation, Helsinki, Finland) inside a 14-plate wind-aspirated radiation shield of model 41005 to measure the $T$ under conditions of potentially significant solar contamination during the day due to low wind-aspiration efficiency. The sensing centers of all sensors related to $T_s$, $T$, and $RH$ were set at height of 2.57 m above the

ground level. The land surface was covered by natural prairie with grass height of 5 to 35 cm.

A CR6, supported by EasyFlux_CR6CP (Revised version for this study, Campbell Scientific Inc. UT, USA), controlled and sampled the EC100 at 20 Hz. For spectral analysis, the EC100 filtered the data of $T_s$ and $\chi_{H2O}$ for anti-aliasing using a finite impulse response filter with a 0-to-10 Hz (Nyquist folding frequency) passing band (Saramäki, 1993). The EC155 was zeroed for $CO_2/H_2O$ and spanned for $CO_2$ automatically every other day and spanned for $H_2O$ monthly using LI-610 Portable Dew point

Generator (LI-Cor Biosciences, Lincoln, NE, US). The LI200, RDT, and HMP155A sampled at 1 Hz because of their slow response and the fact that only their measurement means were of interest to this study.

The purpose of this station was to measure the eddy-covariance fluxes to determine turbulent transfers in the boundary-layer flows. The air temperature equation (i.e. Eq. 23) was developed for $T$ of the air flows sampled through the CPEC systems. Therefore, this equation can be tested based on how the CPEC310 measures the boundary-layer flows related to turbulent

transfer.

## 6 Turbulent transfer and CPEC310 measurement

In atmospheric boundary-layer flows, air constituents along with heat and momentum (i.e. air properties) are transferred dominantly by individual turbulent flow eddies with various sizes (Kaimal and Finnigan, 1994). Any air property is considered as more homogenous inside each smaller eddy and as more heterogenous among larger eddies (Stull, 1988). Due to this

heterogeneity, an eddy in motion among others is transferring air properties to its surroundings. Therefore, to measure the transfer in amount and direction, a CPEC system was designed to capture $T_s$, $\chi_{H_2O}$, and 3-D flow speeds from individual eddies. Ideal measurements should be fast enough to capture, even impossible, all eddies with different sizes through the measurement volume and sampling orifice of the CPEC system (Fig. 1). To capture more eddies, with as many sizes as possible, the CPEC measurements were set at a high frequency (20 Hz in this study) because, given 3-D speeds, the smaller an eddy, the shorter time

the eddy takes to pass the sensor measurement volume.

Ideally, each measurement captures an individual eddy for all variables of interest so that the measured values are representative of this eddy. So, for instance, in our case to compute $T$ from a pair of $T_s$ and $\chi_{H_2O}$, the pair simultaneously measured from the same eddy could better reflect its $T$ at the measurement time; however, in a CPEC system, $T_s$ and $\chi_{H_2O}$ are measured with separation in both space (Fig. 1) and time (Fig. 3).

If an eddy passing the sonic anemometer is significantly larger than the dimension of separation between the $T_s$ measurement volume and the $\chi_{H_2O}$ sampling orifice (Fig. 1), the eddy is instantaneously measured for its 3-D wind and $T_s$ in the volume while also sampled into the orifice for $\chi_{H_2O}$ measurements. However, if the eddy is smaller and flows along the alignment of separation, the sampling takes place either a little earlier or later than the measurement, earlier if $T_s$ is measured later, and vice





370 versa. However, depending on its size, an eddy flowing beyond the alignment from other directions, although measured by the sonic anemometer, may be missed by the sampling orifice passed by other eddies and, in other cases, although sampled by the orifice, may be missed by the measurement of sonic anemometer.

Additionally, the air flow sampled for $\chi_{H_2O}$ measurements is not measured at its sampling time on the sampling orifice, but instead is measured, in lag, inside the $\chi_{H_2O}$ measurement cuvette (Fig. 3). The lag depends on the time needed for the sampled flow to travel through the CPEC sampling system (Fig 3). Therefore, for the computation of $T$, $\chi_{H_2O}$ is better synchronized and
375 matched with $T_s$ as if simultaneously measured from the same eddy.

## 7 Temporal synchronization and spatial match for Ts with $\chi_{H_2O}$

In the CPEC310 system, a pair of $T_s$ and $\chi_{H_2O}$ that were received by CR6 from EC100 in one data record (i.e. data row) were synchronously measured, through Synchronous Device for Measurement Communication Protocol (Campbell Scientific Inc., 2018c), at the same time in the $T_s$ measurement volume and $\chi_{H_2O}$ measurement cuvette (Fig. 1). Accordingly, within one data
380 row of time series received by CR6, $\chi_{H_2O}$ was sampled earlier than $T_s$ was measured. As discussed above, $T_s$ and $\chi_{H_2O}$ in the same row might be measured not from the same eddy although measured at the same time. If so, the $\chi_{H_2O}$ measurement from the same eddy of this $T_s$ might occur in another data row, and vice versa. In any case, a logical procedure for the synchronization match is first to pair $T_s$ with $\chi_{H_2O}$ programmtically in CR6 as the former was measured at the same time as the latter was sampled.

385 **7.1 Synchronize $T_s$ measured to $\chi_{H_2O}$ sampled at the same time**

Among the rows in time series received by CR6, any two consecutive rows were measured sequentially at a fixed time interval (i.e. measurement interval). Accordingly, anemometer data in any data row can be synchronized with analyzer data in a later row from the eddy sampled by the analyzer sampling orifice at the measurement time of sonic anemometer. How many rows later depends on the measurement interval and the time length of the analyzer sample from its sampling orifice to the measurement
390 cuvette. The measurement interval commonly is 50 or 100 ms for 20- or 10-Hz measurement frequency, respectively. The time length is determined by the internal space volume of sampling system (Fig. 3) and the flow rate of sampled air driven by a diaphragm pump (Campbell Scientific Inc., 2018a).

As shown in Fig. 3, the total internal space is 10.563 mL. The rate of sampled air through the sampling system nominally is 6.0 L min$^{-1}$ at which the sampled air takes 106 ms to travel from the analyzer sampling orifice to the cuvette exhaust outlet (Fig.
395 3). Given that the internal optical volume inside the cuvette is 5.887 mL, the air in the cuvette was sampled in a period of 47 to 106 ms earlier. Accordingly, anemometer data in a current row of time series should be synchronized with analyzer data in the next row for 10-Hz data and, for 20-Hz data, the row following next one. After synchronization, the CR6 stores anemometer and analyzer data in a synchronized matrix (variables unrelated to this study were omitted) as a time series:





$$\begin{bmatrix} \cdots\cdots\cdots\cdots \\ u(t_i) \quad v(t_i) \quad w(t_i) \quad T_s(t_i) \quad d_s(t_i) \quad \chi_{H_2O}(t_i) \quad d_g(t_i) \quad s(t_i) \\ \cdots\cdots\cdots\cdots \end{bmatrix} \tag{28}$$

where $u$ and $v$ are horizontal wind speeds orthogonal each other, $w$ is vertical wind speed; $d_s$ and $d_g$ are diagnosis codes for sonic anemometer and gas analyzer, respectively; $s$ is signal strength of gas analyzer for $H_2O$; $t$ is time and its subscript $i$ is its index; and the difference between $t_i$ and $t_{i+1}$ is a measurement interval ($\Delta t = t_{i+1} - t_i$). In any row of matrix (28) (e.g. the $i$th row), $t_i$ for anemometer data is the measurement time plus instrument lag and for analyzer data is the sampling time plus the same lag. The instrument lag is defined as the number of measurement intervals used for data processing inside EC100 after the

measurement and subsequent data communication to CR6. Regardless of instrument lag, $T_s$ and $\chi_{H_2O}$ in each row of synchronization matrix were temporally synchronized as measured and sampled at the same time.

### 7.2 Match $T_s$ measured to $\chi_{H_2O}$ sampled from the same eddy

As discussed in section 6, at either $T_s$ measurement or $\chi_{H_2O}$ sampling time, if an eddy was large enough to enclose both $T_s$ measurement volume and $\chi_{H_2O}$ sampling orifice (Fig. 1), $T_s$ and $\chi_{H_2O}$ in the same row of synchronization matrix (28) belong

to the same eddy, otherwise, to different ones. For any eddy size, it would be ideal if $T_s$ could be spatially matched with $\chi_{H_2O}$ as a pair for the same eddy; however, this match would not be possible for all $T_s$ values simply because, in some cases, an eddy measured by the sonic anemometer might be never sampled by the $\chi_{H_2O}$ sampling orifice, and vice versa (see Section 6). Realistically, $T_s$ may be matched with $\chi_{H_2O}$ overall with most likelihood as many pairs as possible for a period (e.g. an averaging interval).

The match is eventually to lag either $T_s$ or $\chi_{H_2O}$, relatively, in the synchronization matrix. The lag can be counted as an integer number ($l_s$, subscript $s$ indicates the spatial separation causing lag) in measurement intervals where $l_s$ is positive if an eddy flowed through the $T_s$ measurement volume earlier, negative if later, or zero if through the $\chi_{H_2O}$ sampling orifice at the same time. This number is estimated through the covariance maximization (Irwin, 1979; Moncrieff et al., 1997; Ibrome et al., 2007; Rebmann et al., 2012). According to $l_s$ over an averaging interval, the data columns of the gas analyzer over an averaging

interval in synchronization matrix (28) can be moved together up $l_s$ rows as positive, down $l_s$ rows as negative, or nowhere as zero to form a matched matrix:

$$\begin{bmatrix} \cdots\cdots\cdots\cdots \\ u(t_i) \quad v(t_i) \quad w(t_i) \quad T_s(t_i) \quad d_s(t_i) \quad \chi_{H_2O}(t_{i+l_s}) \quad d_g(t_{i+l_s}) \quad s(t_{i+l_s}) \\ \cdots\cdots\cdots\cdots \end{bmatrix} \tag{29}$$

For details to find $l_s$, see EasyFlux_CR6CP on https://www.campbellsci.com. In the matched matrix, over an averaging interval, a pair of $T_s$ and $\chi_{H_2O}$ in the same row can be assumed to be matched as if measured and sampled from the same eddy.

Using Eq. (23), the air temperature now can be computed using:





$$T_{l_s,i} = T_s(t_i) \frac{[1 + \varepsilon \chi_{H_2O}(t_{i+l_s})][1 + \varepsilon \gamma_v \chi_{H_2O}(t_{i+l_s})]}{[1 + \chi_{H_2O}(t_{i+l_s})][1 + \varepsilon \gamma_p \chi_{H_2O}(t_{i+l_s})]} \qquad (30)$$

where subscript $l_s$ for $T$ indicates that spatially lagged $\chi_{H_2O}$ is used for computation of $T$. Verification for the accuracy of equation-computed $T$ and in assessments on its expected advantages of high frequency signal insensitive to solar contamination in measurements, $T_{l_s,i}$ could minimize the uncertainties due to the spatial separation in measurements of $T_s$ and $\chi_{H_2O}$ between

the $T_s$ measurement volume and the $\chi_{H_2O}$ sampling orifice (Fig. 1).

## 8 Verification for the accuracy of equation-computed $T$

The accuracy of equation-computed $T$ was theoretically specified by Eqs. (25) to (27) and was estimated in Fig. 2c. This accuracy specifies the range of equation-computed minus true $T$ (i.e. $\Delta T$). However, the true $T$ was not available in the field but, as usual, precision measurements could be considered as a benchmark to represent the true $T$. In this study, $T$ measured by the

RTD inside a fan-aspirated radiation shield ($T_{RTD}$) was the benchmark to compute $\Delta T$ (i.e. equation-computed $T$ minus $T_{RTD}$). If almost all $\Delta T$ values fall within the accuracy-specified range over a measurement domain of $T_c$ and $\chi_{H_2O}$, the accuracy is correctly defined and the equation-computed $T$ is accurate as specified.

To verify the accuracy over the domain as large as possible, $\Delta T$ values in the coldest (January) and hottest (July) months were used as shown in Fig. 4 (-21 °C < $T_c$ < 35.5 °C and $\chi_{H_2O}$ up to 20.78 mmolH$_2$O mol$^{-1}$ in a 30-minute mean over the two months).

Out of 2,976 $\Delta T$ values from both months, 44 values fell out of specified-accuracy range, but near the range line within 0.30 K. The $\Delta T$ values were 0.549 ± 0.281 K in January and 0.436 ± 0.290 K in July. Although these values were almost all positively away from the zero-line due to either overestimation for $T_s$ by the sonic anemometer within ± 1.00 K accuracy or underestimation for $T_{RTD}$ by the RTD within ± 0.20 K accuracy, the ranges are significantly narrower than the specified accuracy range of equation-computed $T$ (Figs. 2c and 4). Therefore, the equation-computed $T$ is accurate as specified and even much

better.





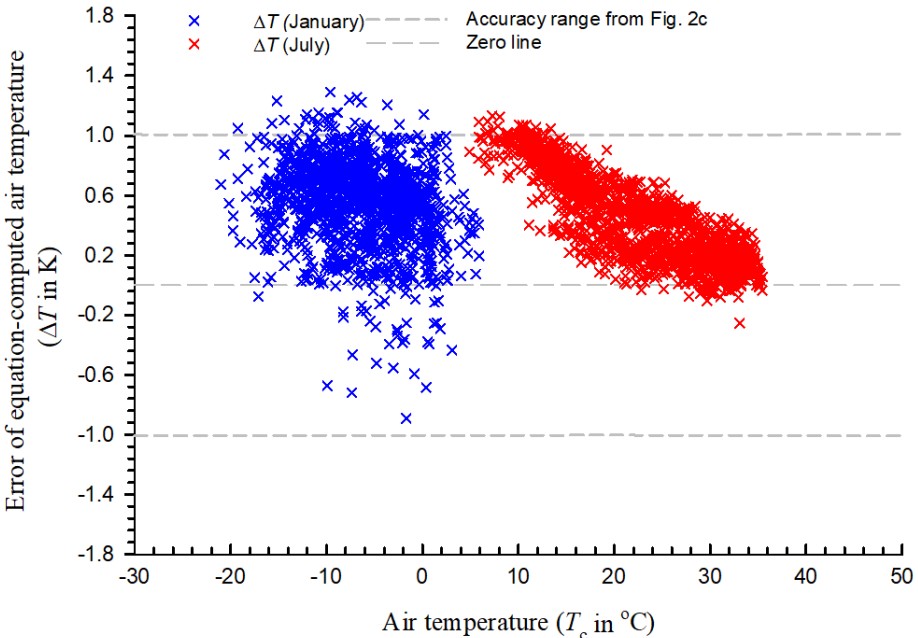

**Figure 4: The error of equation-computed air temperature in coldest (January) and hottest (July) months, 2019 in Logan, UT, US. $\Delta T$ is equation-computed minus RTD-measured air temperature where RTD is a precision platinum resistance temperature detector inside a fan-aspirated radiation shield. $\Delta T$: 0.549 ± 0.281 K in January and 0.436 ± 0.290 K in July. See Fig. 2c for the accuracy range.**

**9 Assessments of the advantages of equation-computed $T$**

As previously discussed, the data stream of equation-computed $T$ consists of high frequency signals insensitive to solar contamination in measurements. Its frequency response can be assessed against known high frequency signals of $T_s$ and the insensitivity can be assessed by analyzing the equation-computed, RDT-measured, and sensor-measured $T$ where the sensor is HMP155A inside a wind-aspirated radiation shield.

**9.1 Frequency response**

The matched matrix (29) and Eq. (30) were used to compute $T_{l_s,i}$ (i.e. equation-computed $T$). Paired power spectra of equation-computed $T$ and $T_s$ are compared in Fig. 5 for three individual two-hour periods of unstable ($z/L$ = -0.313 ~ -2.999 where $z$ is a dynamic height of measurement minus displacement height and $L$ is the Monin-Obukhov length), near- neutral ($z/L$ = -0.029 ~ +0.003), and stable ($z/L$ = +0.166 ~ +0.600) atmospheric stratifications. Slower response of equation-computed $T$ than $T_s$ at

higher frequency (e.g. > 5 Hz) was expected because equation-computed $T$ is derived from two variables ($T_s$ and $\chi_{H_2O}$) measured in a spatial separation, which attenuates the frequency response of correlation of two measured variables (Laubach and McNaughton, 1998), and $\chi_{H_2O}$ from a CPEC system has slower response than $T_s$ in frequency (Ibrom et al., 2007). However, the expected slower response was not found in this study. In unstable and stable atmospheric stratifications (panels a and c of Fig. 5), each pair of power spectra almost overlap. Although they do not overlap in the near-neutral atmospheric stratification, the





pair follow the same trend slightly above or below one another. In the higher frequency band of 1 to 10 Hz in panels a and b of Fig 5, equation-computed $T$ has a little more power than $T_s$. The three pairs of power spectra in Fig. 5 indicate that equation-computed $T$ has frequency reponse equalvelent to $T_s$ up to 10 Hz at 20-Hz measurent rate considered as a high frequency. The equivalent response might be accounted for by a dominant role of $T_s$ in the magnitude of equation-computed $T$.

**Figure 5: Paired comparisons of power spectra for equation-computed air temperature and sonic temperature ($T_s$) at each of atmospheric stratifications: unstable (a), near-neutral (b), and stable (c). $T_1$ and $T_{-3}$ are equation-computed air temperature from $T_s$**



and the H₂O mixing ratio of air sampled by the CPEC system through its sampling orifice in 50 ms behind (1 lag) and in 150 ms ahead (-3 lags) of $T_s$ measurement; $z$ is the dynamic height of measurement minus displacement height, $L$ is Monin-Obukhov length, $S_{T_s}(f)$, $S_{T_1}(f)$ and $S_{T_{-3}}(f)$ are the power spctra of $T_s$, $T_1$, and $T_{-3}$ at $f$; and $\sigma^2_{T_s}$, $\sigma^2_{T_1}$ and $\sigma^2_{T_{-3}}$ are the variance of $T_s$, $T_1$, and $T_{-3}$.

## 9.2 Insensitivity to solar contamination in measurements

The data of equation-computed, sensor-measured, and RDT-measured $T$ in July, during which incoming solar radiation ($R_s$) in the site was strongest in a yearly cycle, were used to assess the sensitivity of equation-computed $T$. From the data, $\Delta T$ is considered as an error of equation-computed $T$. The error of sensor-measured $T$ can be defined as sensor-measured minus RDT-measured $T$, denoted by $\Delta T_m$. From Fig. 6, $\Delta T$ (0.690 ±0.191 K) is > $\Delta T_m$ (0.037±0.199 K) when $R_s$ < 50 W m⁻² at lower radiation. However, $\Delta T$ (0.234 ±0.172 K) is < $\Delta T_m$ (0.438 ±0.207 K) when $R_s$ > 50 W m⁻² at higher radiation. This difference between $\Delta T$ and $\Delta T_m$ shows a different effect of $R_s$ on equation-computed and sensor-measured $T$.

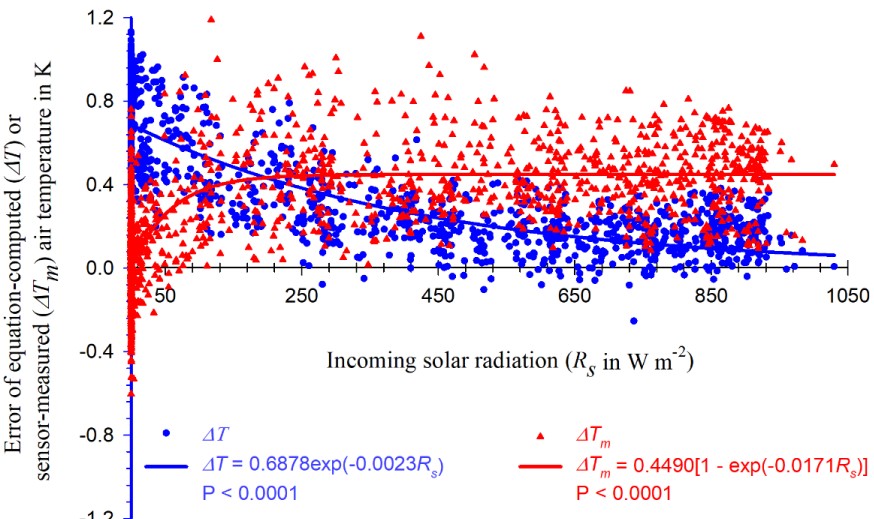

**Figure 6: Errors in equation-computed and sensor-measured air temperature with incoming solar radiation. $\Delta T$ is equation-computed minus RDT-measured air temperature where the RDT is a precision platinum resistance temperature detector inside a fan-aspirated radiation shield. $\Delta T_m$ is sensor-measured minus RDT-measured air temperature where the sensor is a HMP155A air temperature and humidity probe inside a wind-aspirated radiation shield.**

As shown from Fig. 6, $\Delta T_m$ increases sharply with increasing $R_s$ for $R_s$ < 250 W m⁻², beyond which it asymptotically approaches 0.40 K. In the range of lower $R_s$, atmospheric stratification was likely stable (Kaimal and Finnigan, 1994) under which the heat exchange by wind was ineffective between the wind-aspirated radiation shield and boundary-layer flows. In this case, sensor-measured $T$ was expected to increase with $R_s$ increase (Lin et al., 2001; Blonquist and Bugbee, 2018). Along with $R_s$ increase, the atmospheric boundary-layer develops from stable to neutral or unstable conditions (Kaimal and Finnigan, 1994). During the stability change, the exchange becomes increasingly more effective, offsetting the further heating from $R_s$ increase on the wind-aspirated radiation shield as indicated by the red asymptote portion in Fig. 6. Compared to $\Delta T_m$ mean (0.037 K) while $R_s$ < 50 W m⁻², the magnitude of the asymptote above the mean is the over-estimation of sensor-measured $T$ due to solar contamination.

However, $\Delta T$ decreases asymptotically from about 0.70 K toward zero with the increase in $R_s$ from 50 to 250 W m⁻² and beyond, with a more gradual rate of change than $\Delta T_m$ at the lower radiation range. Lower $R_s$ (e.g. < 250 W m⁻²) concurrently





occurs with lower $T_c$, higher RH, and/or unfavorable weather to $T_s$ measurements. Under lower $T_c$ (e.g. below 20 ºC of CSAT3A manufacture conditions), the sonic path lengths of CSAT3A (Fig. 1) must become, due to thermo-contraction of sonic

anemometer structure, shorter than these at 20 ºC. As a result, the sonic anemometer could over-estimate the speed of sound (Zhou et al., 2018) and hence $T_s$ for equation-computed $T$, resulting in greater $\Delta T$ with lower $R_s$. Under higher RH conditions, dew may form on the sensing surface of CSAT3A six sonic transducers (Fig. 1). The dew, along with unfavorable weather, could contaminate the $T_s$ measurements, resulting in greater $\Delta T$ in magnitude. Higher $R_s$ (e.g. > 250 W m$^{-2}$) concurrently occurs with weather favorable to $T_s$ measurements, which is the reason that $\Delta T$ slightly decreases rather than increases with $R_s$ when $R_s > 250$

W m$^{-2}$.

Again from Fig. 6, the data pattern of $\Delta T > \Delta T_m$ in the lower $R_s$ range and $\Delta T < \Delta T_m$ in the higher $R_s$ range shows that equation-computed $T$ is not sensitive to $R_s$ as sensor-measured $T$. The decreasing trend of $\Delta T$ with $R_s$ increase shows the insensitivity of equation-computed $T$ to $R_s$.

**10 Discussion**

**10.1 Actual accuracy**

The range of $\Delta T$ curves for each *RH* level in Fig. 2 is the maximum at that level because the data were evaluated using the maximized measurement uncertainties from all sources. Accordingly, in field applications under weather favorable to $T_s$ measurements, the range of actual accuracy in equation-computed $T$ can be reasonably inferred to be narrower. In our study case as shown in Figs. 4 and 6, the variability of $\Delta T$ was narrower than the accuracy range as specified in Fig. 2. The actual accuracy

is better.

However, under weather conditions unfavorable to $T_s$ measurements such as dew, rain, snow, or dust storm, the accuracy of $T_s$ measurements cannot be easily evaluated. $T_s$ measurements also possibly have a systematic error due to the fixed deviation in the measurements of sonic path lengths for sonic anemometers although the error should be within the accuracy specified in Fig. 2.

A $\chi_{H_2O}$ measurement also can be erroneous if the gas analyzer is not periodically zeroed and spanned for its measurement

environment. Therefore, if $T_s$ is measured under unfavorable weather conditions, the sonic anemometer produces a systematic $T_s$ error, and the gas analyzer is not zeroed and spanned as instructed in its manual; the accuracy of equation-computed $T$ would be unpredictable. Normally, the actual accuracy is better than the one specified in Fig. 2. Additionally, with the improvement in measurement accuracies of sonic anemometers [e.g. Weather-condition-regulated, heated 3-D sonic anemometers, Mahan et al. (2021)] and gas analyzers, this accuracy of equation-computed $T$ would be gradually becoming better and better.

**10.2 Spatial separation of $T_s$ and $\chi_{H_2O}$ in measurements**

In this study, $T$ was successfully computed from the $T_s$ and $\chi_{H_2O}$ as a high- frequency signal with expected accuracy as tested in Figs. 4 and 6, where both were measured separately from two sensors in a spatial separation. Some OPEC flux systems (e.g. CSAT3A+EC150 and CSAT3B+LI7500) measure $T_s$ and $\rho_w$ also from two sensors in a spatial separation. To an OPEC system, although the air temperature equation (Eq. 23) is not applicable, the algorithms developed in Section 7 to temporally synchronize

and spatially match $T_s$ with $\chi_{H_2O}$ for computation of $T$ are applicable for computation of $T$ from $T_s$ and $\rho_w$ along with $P$ in such OPEC systems (Swiatek, 2018).





In Section 7, programing and computing are needed to pair $T_s$ measured to $\chi_{H_2O}$ sampled at the same time into synchronization matrix (28) as the first step and from the same eddy into matched matrix (29) as the second step. The second requires complicated programing and much computing. To test the necessity of this step in specific cases, using Eq.

(30), $T_{0i}$ was computed from a row of the synchronization matrix and $T_{l_s,i}$ was computed from this matrix by lagging $\chi_{H_2O}$ columns up $l_s$ rows if $l_s > 0$ and down $|l_s|$ rows if $l_s < 0$ where $l_s$ is -5, ……, -1, +1, ……, +5. From the data of this study, individual $T_{l_s,i}$ values were different for different subscript $l_s$, but their means for subscript $i$ over an averaging interval $(T_{l_s})$ are the same to at least the fourth digit after the decimal place. Further, the power spectrum of $T_{0i}$ time series was compared to those of $T_{l_s,i}$ time series, where $i \neq 0$. Any pair of power spectra from the same period overlap exactly (Figures omitted). Therfore, the

second step of lag maximization to match $T_s$ measured to $\chi_{H_2O}$ sampled from the same eddy is not needed if only hourly mean and power spectrum of equation-computed $T$ are of interest to computations, for both CPEC and OPEC systems.

### 10.3 Applications

The air temperature equation (Eq. 23) is derived from first principles without any assumption and approximation. It is an exact equation from which $T$ can be computed in a CPEC system as a high frequency signal insensitive to solar radiation. Therefore,

this equation is applicable to calculations of $\rho_d$ in Eq. (1), sensible heat flux from $T_s$ in a possible way different from Schotanus et al. (1983) or van Dijk (2002), and RH as a high frequency signal in a CPEC system.

#### 10.3.1 Dry air density

As a high frequency signal insensitive to solar radiation, equation-computed $T$ is more applicable than sensor-measured $T$ for calculations of $\overline{\rho}_d$ and $\overline{\rho_d w}$ for more applications (Gu et al., 2012; Foken et al., 2012). In practice, equation-computed $T$ may

be used for $\overline{\rho}_d$ and $\overline{\rho_d w}$ under normal weather conditions while sonic anemometer and gas analyzer are normally running, which can be judged by their diagnosis codes (Campbell Scientific Inc., 2018a). Under a weather condition unfavorable to $T_s$ measurements due to dew, rain, snow, and ice conditions; equation-computed $T$ from weather-condition-regulated, heated 3-D sonic anemometers (Mahan et al., 2021) and CPEC gas analyzer could be an alternative.

#### 10.3.2 Sensible heat flux estimated from an CPEC system

Currently, sensible heat flux ($H$) is derived from $\overline{T_s' w'}$ with a humidity correction (van Dijk, 2002). The correction equations were derived by Schotanus et al. (1983) and van Dijk (2002) in two ways, but both from Eq. (4) derived with approximation (see Appendix A). Using the exact equation from this study, theoretically, $H$ can be more accurately estimated directly from $\overline{T' w'}$, where $T$ is the equation-computed air temperature, although studies and tests for this potential application are needed.

#### 10.3.3 RH as a high frequency signal

Conventionally, RH is measured using a temperature and humidity probe which is unable to track the high frequency fluctuations





of RH. In a CPEC system, equation-computed $T$, gas-analyzer-measured $\chi_{H_2O}$, and transducer-measured $P$ are able to catch the fluctuations in these variables at high frequency, from which RH can be computed (Sonntag, 1990; also see Appendix C). This method should provide high frequency RH although verification is needed. Currently, the applications of high frequency properties in this RH are unknown in a CPEC system. Regardless, equation-computed $T$ provides a a potential opportunity to acquire the high frequency RH for its application in the future.

## 11 Concluding remarks

In a CPEC flux system, the air temperatrure ($T$) of boundary-layer flows through the space of sonic anemometer measurement and gas analyzer sampling (Fig. 1) is desired for high frequency (e.g. 10 Hz) with consistent representation of spatial and temporal scales for moist turbulence thermodynamics characterized by three-dimensional wind from the sonic anemometer and $H_2O/CO_2$ and atmospheric pressure from the gas analyzer measurements. High frequency $T$ in the space can be measured using fine wire thermocouples, but this kind of thermocouples for such an application is not durable under adverse climate conditions, being easily contaminated by solar radiation (Campbell, 1969). Nevertheless, the measurements of sonic temperature ($T_s$) and $H_2O$ inside a CPEC system are high frequency signals. Therefore, high frequency $T$ can be reasonably expected when computed from $T_s$ and $H_2O$-related variables. For this expectation, two equations [Eqs. (4) and (5)] are currently available. In both equations, converting $H_2O$-related variables into $H_2O$ mixing ratio analytically reveals the difference between the two equations. This difference in CPEC systems reaches ±0.18 K, bringing an uncertainty into the accuracy of $T$ from either equation and raising a question of which equation is better. To clarify the uncertainty and answer this question, the air temperature equations in terms of $T_s$ and $H_2O$-related variables are thoroughly reviewed (Sections 2 and 3, Appendices A and B). The two currently used equations [i.e. Eqs. (4) and (5)] were developed and completed with approximations (Appendices A and B). Because of the approximations, neither of their accuracies were evaluated, nor was the question answered.

Using the first principles equations, the air temperature equation in terms of $T_s$ and $\chi_{H_2O}$ ($H_2O$ mixing ratio) is derived without any assumption and approximation (Eq. 23); therefore, the equation derived in this study does not, itself, have any error and, as such, the accuracy in equation-computed $T$ depends solely on the measurement accuracies of $T_s$ and $\chi_{H_2O}$. Based on the specifications for $T_s$ and $\chi_{H_2O}$ in the CPEC300 series, the accuracy of equation-computed $T$ over the $T_s$ and $\chi_{H_2O}$ measurement ranges can be specified within ±1.01K (Fig. 2). This accuracy uncertainty is propagated mainly (±1.00K) from the uncertainty in $T_s$ measurements (Fig. 2a) and little (±0.03 K) from the uncertainty in $\chi_{H_2O}$ measurements (Fig. 2b).

Under normal sensor and weather conditions, the specified accuracy is verified based on field data as valid and actual accuracy is better (Figs. 4 and 6). Field data demonstrated that equation-computed $T$ under unstable, near-neutral, and stable atmospheric stratifications all have frequency responses equalvelent to high-frequency $T_s$ up to 10 Hz at 20-Hz measurement rate (Fig. 5), being insensitive to solar contamination in measurements (Fig. 6).

The current applications of equation-computed $T$ in a CPEC system are to calculate $\rho_d$ for the estimations of $CO_2$ flux ($\overline{\rho_d\,\chi'_{CO_2}w'}$, where $\chi_{CO_2}$ is $CO_2$ mixing ratio), $H_2O$ flux ($\overline{\rho_d\,\chi'_{H_2O}w'}$), and other fluxes. Combined with measurements of $\chi_{H_2O}$ and $P$, the equation-computed $T$ can be potentially applied to the computation of high frequency RH (Sonntag, 1990) and to the estimation of sensible heat flux ($H$) avoiding the humidity correction as needed for $H$ indirectly from $T_s$ (Schotanus et





al., 1983; van Dijk, 2002)

In a CPEC flux system, although $T_s$ and $\chi_{H_2O}$ are measured using two spatially separated sensors of sonic anemometer and gas analyzer, $T$ was successfully computed from both measured variables as a high-frequency signal (Fig. 5) with an expected accuracy (Figs. 2 and 4). Some OPEC flux systems measure $T_s$ and water vapor density ($\rho_w$) also from two sensors in a similar way. The algorithms developed in Section 7 to temporally synchronize and spatially match $T_s$ with $\chi_{H_2O}$ for computation of $T$

are applicable to such an OPEC system to compute $T$ from $T_s$ and $\rho_w$ along with $P$. This $T$ would be a better option than sensor-measured $T$ in the system for the correction of spectroscopic effect in measuring $CO_2$ fluctuations at high frequencies (Helbig et al., 2016; Wang et al., 2016). With the improvements on measurement technologies for $T_s$ and $\chi_{H_2O}$, particularly for $T_s$, the $T$ from our developed equation will become increasingly more accurate. Having its accuracy combined with its high frequency, this $T$ with consistent representation of all other thermodynamic variables for moist air at the spatial and temporal scales in

CPEC measurements has its advanced merits in boundary-layer meteorology and applied meteorology.

**Appendices**

**Appendix A. Derivation of equation (4)**

The sonic temperature ($T_s$) reported by a three-dimensional sonic anemometer-thermometry is internally calculated from its

measurements of the speed of sound in moist air ($c$) after the crosswind correction (Zhou et al., 2018), using

$$T_s = \frac{c^2}{\gamma_d R_d},$$ (a1)

where subscript $d$ indicates dry air, $\gamma_d$ is the specific heat ratio of dry air between constant pressure and constant volume, and $R_d$ is gas constant for dry air (Campbell Scientific Inc., 2018b). The speed of sound in the atmospheric boundary-layer as in a homogeneous gaseous medium is well defined in acoustics (Barrett and Suomi, 1949), given by:

$$c^2 = \gamma \frac{P}{\rho},$$ (a2)

where, as the counterpart of $\gamma$ is the specific heat ratio of moist air; $P$ is atmospheric pressure; and $\rho$ is moist air density. These variables are related to air temperature and air specific humidity ($q$, i.e. the mass ratio of water vapor to moist air).

**1. Moist air density ($\rho$)**

Moist air density is the sum of dry air and water vapor densities. Based on the ideal gas law (Wallace and Hobbs, 2006), dry air

density ($\rho_d$) is given by:

$$\rho_d = \frac{P - e}{R_d T},$$ (a3)

where $e$ is water vapor pressure, and the water vapor density ($\rho_w$) is given by:

$$\rho_w = \frac{e}{R_v T},$$ (a4)

where $R_v$ is the gas constant for water vapor. Therefore, moist air density in Eq. (a2) can be expressed as





$$\rho = \frac{P-e}{R_d T} + \frac{e}{R_v T} .$$
(a5)

Because of $R_d/R_v = \varepsilon$ (i.e. 0.622, the molar mass ratio between water vapor and dry air), this equation can be rearranged as:

$$\rho = \frac{P}{R_d T}\left[1-(1-\varepsilon)\frac{e}{P}\right].$$
(a6)

Using Eqs. (a4) and (a6), the air specific humidity can be expressed as

$$q \equiv \frac{\rho_w}{\rho} = \frac{\varepsilon e}{P-(1-\varepsilon)e} .$$
(a7)

Because of $P \gg (1-\varepsilon)e$, $q$ can be approximated as

$$q \approx \varepsilon \frac{e}{P} .$$
(a8)

Substituting this relation into Eq. (a6) generates:

$$\rho = \frac{P}{R_d T}\left(1-\frac{1-\varepsilon}{\varepsilon}q\right).$$
(a9)

**2. Specific heat ratio of moist air** ($\gamma$)

The specific heat ratio of moist air is determined by two moist air properties: the specific heat at constant pressure ($C_p$) and specific heat at constant volume ($C_v$). $C_p$ varies with the air moisture content between the specific heat of dry air at constant pressure ($C_{pd}$) and the specific heat of water vapor at constant pressure ($C_{pw}$). It must be the average of $C_{pd}$ and $C_{pw}$ that are arithmetically weighted by dry air mass and water vapor mass, respectively, given by (Stull, 1988):

$$C_p = \frac{C_{pd}\rho_d + C_{pw}\rho_w}{\rho} .$$
(a10)

$C_v$ can be similarly determined:

$$C_v = \frac{C_{vd}\rho_d + C_{vw}\rho_w}{\rho} ,$$
(a11)

where $C_{vd}$ is the specific heat of dry air at constant volume and $C_{vw}$ is the specific heat of water vapor at constant volume. Denoting $C_{pd}/C_{vd}$ as $\gamma_d$, Eqs. (a10) and (a11) are used to express $\gamma$ as:

$$\gamma = \frac{C_p}{C_v} = \gamma_d \frac{(1-q)+qC_{pw}/C_{pd}}{(1-q)+qC_{vw}/C_{vd}} .$$
(a12)

**3. Relate sonic temperature to air temperature**

Substituting Eqs. (a9) and (a12) into Eq. (a2) leads to:

$$c^2 = \gamma_d R_d T \frac{(1-q)+qC_{pw}/C_{pd}}{\left[(1-q)+qC_{vw}/C_{vd}\right]\left(1-\frac{1-\varepsilon}{\varepsilon}q\right)} .$$
(a13)

Using this equation to replace $c^2$ in Eq. (a1), $T_s$ is expressed as

$$T_s = T \frac{(1-q)+qC_{pw}/C_{pd}}{\left[(1-q)+qC_{vw}/C_{vd}\right]\left(1-\frac{1-\varepsilon}{\varepsilon}q\right)} .$$
(a14)





Given $C_{pw} = 1,952$, $C_{pd} = 1,004$, $C_{vw} = 1,463$, and $C_{vd} = 717$ J K$^{-1}$ kg$^{-1}$ (Wallace and Hobbs, 2006); this equation becomes:

$$T_s = T(1+0.944223q)\left(\frac{1}{1+1.040446q}\right)\left(\frac{1}{1-0.607717q}\right).$$ (a15)

Expression of the last two parenthesized terms in the right side of this equation separately as Taylor series of $q$ (Burden and Faires, 1993) by dropping, due to $q \ll 1$, the second or higher terms related to $q$ leads to

$$T_s \approx T(1+0.944223q)(1-1.040446q)(1+0.607717q).$$ (a16)

In the right side of this equation, the three parenthesized terms can be expanded into a polynomial of $q$ at the third order. Also due to $q \ll 1$ in this polynomial, the terms of $q$ at the second or third order can be dropped. Further arithmetical manipulations result in:

$$T_s \approx T(1+0.51q).$$ (a17)

This is Eq. (4) in a different form. In its derivations from Eqs. (a1) and (a2), three approximation procedures were used from Eq.
(a7) to (a8), (a15) to (a16), and (a16) to (a17). The three approximations must bring unspecified errors into the derived equation.

**Appendix B. Derivation of equation (5)**

Equation (5) was sourced from Ishii (1932) in which the speed of sound in moist air ($c$) was expressed in his Eq. (1) as:

$$c^2 = \gamma\left(\frac{P}{\rho}\right)\left(\frac{\alpha}{\beta}\right),$$ (b1)

where all variables in this equation are for moist air, $\gamma$ is the specific heat ratio of moist air between constant pressure and
constant volume, $P$ is moist air pressure, $\rho$ is moist air density, $\alpha$ is moist air expansion coefficient, and $\beta$ is moist air pressure coefficient. Accordingly, the speed of sound in dry air ($c_d$) is given by:

$$c_d^2 = \gamma_d\left(\frac{P_d}{\rho_d}\right)\left(\frac{\alpha_d}{\beta_d}\right),$$ (b2)

where subscript $d$ indicates dry air in which $\gamma_d$, $P_d$, $\rho_d$, $\alpha_d$, and $\beta_d$ are the counterparts of $\gamma$, $P$, $\rho$, $\alpha$, and $\beta$ in moist air. Equations (b1) and (b2) can be combined as

$$c^2 = c_d^2\left(\frac{\gamma}{\gamma_d}\right)\left(\frac{P\rho_d}{P_d\rho}\right)\left(\frac{\alpha\beta_d}{\alpha_d\beta}\right).$$ (b3)

Experimentally by Ishii (1932), each term inside the three pairs of parentheses in this equation was linearly related to the ratio of water vapor pressure ($e$) to dry air pressure ($P_d$). The relationship into Eq. (b3) leads to:

$$c^2 = c_d^2\left(1+0.00163\frac{e}{P_d}\right)\left(1-0.378\frac{e}{P_d}\right)^{-1}\left(1-0.0613\frac{e}{P_d}\right).$$ (b4)

The three parenthesized terms in this equation are sequentially corresponding to the three parenthesized terms in Eq. (b3). Diving
$\gamma_d R_d$, where $R_d$ is gas constant for dry air, over both sides of Eq. (b4) and reference Eq. (11), sonic temperature ($T_s$) is expressed in terms of air temperature ($T$), $e$, and $P_d$ as:

$$T_s = T\left(1+0.00163\frac{e}{P_d}\right)\left(1-0.378\frac{e}{P_d}\right)^{-1}\left(1-0.0613\frac{e}{P_d}\right)$$ (b5)



Using the relationship of $P_d = P - e$, Eq. (b5) can be manipulated as:

$$T_s = T\left(\frac{P-0.9984e}{P-e}\right)\left(\frac{P-1.3780e}{P-e}\right)^{-1}\left(\frac{P-1.0613e}{P-e}\right)$$

$$= T\left(\frac{P-0.9984e}{P-e}\right)\left(\frac{P-1.0613e}{P-1.3780e}\right) \qquad (b6)$$

$$= T\frac{1-2.0597e/P+1.0596(e/P)^2}{1-2.3780e/P+1.3780(e/P)^2}$$

Dropping the second order terms due to $e/P \ll 1$ in boundary-layer flows, this equation becomes:

$$T_s \approx T\left(1-2.0597\frac{e}{P}\right)\left(1-2.3780\frac{e}{P}\right)^{-1} \qquad (b7)$$

Expending the second parenthesized term into Taylor series and, also due to $e/P \ll 1$, dropping the terms related to $e/P$ at an order of second or higher, this equation becomes:

$$T_s \approx T\left(1-2.0597\frac{e}{P}\right)\left(1+2.3780\frac{e}{P}\right) \qquad (b8)$$

Further expending the two parenthesized terms in the right side of this equation and dropping the second order term of $e/P$ lead to:

$$T_s \approx T\left(1+0.32\frac{e}{P}\right) \qquad (b9)$$

This is Eq. (5) in a different form. From the experimental source of Eq. (b4), it was derived using three approximations from Eq. (b4) to (b7), (b7) to (b8), and (b8) to (b9). The approximations, combined uncertainty in $T$ therefore, bring unspecified errors into
Eq. (5) [i.e. Eq. (b9)] as an equation error.

**Appendix C. Water vapor mixing ratio and sonic temperature from relative humidity, air temperature, and atmospheric pressure**

For a given air temperature ($T_c$ in °C) and atmospheric pressure ($P$ in kPa), air has a limited capacity to hold water vapor (Wallace and Hobbs, 2006). This limited capacity is described in terms of saturation water vapor pressure ($e_s$ in kPa) for moist
air, given through the Clausius-Clapeyron equation (Sonntag, 1990):

$$e_s(T_c, P) = \begin{cases} 0.6112\exp(\dfrac{17.62T_c}{T_c+243.12})f(P) & T_c \geq 0 \\[4mm] 0.6112\exp(\dfrac{22.46T_c}{T_c+272.62})f(P) & T_c < 0 \end{cases} \qquad (c1)$$

Where $f(P)$ is an enhancement factor for moist air, being a function of atmospheric pressure: $f(P) = 1.0016 + 3.15\times10^{-5}P - 0.0074P^{-1}$. At relative humidity (RH in %), the water vapor pressure



$[\,e_{\mathrm{RH}}(T_c,P)\,]$ is:

$$e_{\mathrm{RH}}(T_c,P) = \frac{\mathrm{RH}}{100} e_s(T_c,P) \tag{c2}$$

Given the mole numbers of $H_2O$ ( $n_{\mathrm{RH}}$ ) and dry air ($n_d$) at RH, the water vapor mixing ratio at RH ( $\chi_{H_2O}^{\mathrm{RH}}$ ):

$$\chi_{H_2O}^{\mathrm{RH}} \equiv \frac{n_{\mathrm{RH}}}{n_d} = \frac{n_{\mathrm{RH}} R^*(T_c + 273.15)}{n_d R^*(T_c + 273.15)} = \frac{e_{\mathrm{RH}}(T_c,P)}{P_d} \tag{c3}$$

where $R^*$ is the universal gas constant and $P_d$ is dry air pressure. Using this equation and the relation:

$$P = P_d + e_{\mathrm{RH}}(T_c,P) \tag{c4}$$

$\chi_{H_2O}^{\mathrm{RH}}$ can be expressed as

$$\chi_{H_2O}^{\mathrm{RH}} = \frac{e_{\mathrm{RH}}(T_c,P)}{P - e_{\mathrm{RH}}(T_c,P)} \tag{c5}$$

Using Eq. (23), this $\chi_{H_2O}^{\mathrm{RH}}$ along with $T_c$ can be used to calculate sonic temperature ($T_s$) at RH, given by:

$$T_s(T_c,\chi_{H_2O}^{\mathrm{RH}}) = (T_c + 273.15)\frac{\left(1+\chi_{H_2O}^{\mathrm{RH}}\right)\left(1+\varepsilon\gamma_p\chi_{H_2O}^{\mathrm{RH}}\right)}{\left(1+\varepsilon\chi_{H_2O}^{\mathrm{RH}}\right)\left(1+\varepsilon\gamma_v\chi_{H_2O}^{\mathrm{RH}}\right)} \tag{c6}$$

where $\varepsilon = 0.622$ (Eq. 17), $\gamma_v = 2.04045$, and $\gamma_p = 1.94422$ (Eq. 23). Through Eqs. (c1) and (c2), Eqs. (c5) and (c6)

express $\chi_{H_2O}^{\mathrm{RH}}$ and $T_s(T_c,\chi_{H_2O}^{\mathrm{RH}})$, respectively, in terms of $T_c$, RH, and $P$. $\chi_{H_2O}^{\mathrm{RH}}$ and $T_s(T_c,\chi_{H_2O}^{\mathrm{RH}})$ can be used to

replace $\chi_{H_2O}$ (water vapor mixing ratio) and $T_s$ in Eq. (25). After replacements, Eq. (25) can be used to evaluate the uncertainty,

due to $T_s$ and $\chi_{H_2O}$ measurement accuracy uncertainties, in air temperature computed from Eq. (23) for different *RH* values over

a $T_c$ range.

**Author Contribution.**

JZ proposes and led this work; XinZ, TG, and XiaoZ derive equations, analyze data, and draft manuscript; ET substantially
structure and revise manuscript; and AS, TA, and JO make comments on the manuscript.

**Competing interest**

The authors declare that they have no conflict of interest.

**Acknowledgments**

Authors thanks Rex Burgon for his advices about the technical design of CPEC sampling system and Edward Swiatek for his
setting the CPEC system in Campbell Scientific Instrument Test Field. This research was supported by the Strategic Priority





Research Program of the Chinese Academy of Sciences (XDA19030204) and Chinese Academy of Sciences President's
International Fellowship Initiative (2020VBA0007).

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
