# Peer review of "Air temperature equation derived from sonic temperature and water vapor mixing ratio for turbulent air flow sampled through closed-path eddy-covariance flux systems"

_Atmospheric Measurement Techniques, 2021_

## Referee Comment (RC2)

The paper is clearly written and makes a valuable scientific contribution with the derivation of equation (23), an equation based on first principles that can be used to compute high frequency air temperature from measurements with closed path eddy covariance systems. As shown in the paper, equation (23) is an advance beyond the previously used approximations, equations (4) and (5). And, as described in the paper, high frequency air temperature computed from equation (23) has potential to improve flux calculations with the eddy covariance technique. The paper should be published, but full consideration should be given the comments below, as there is opportunity to improve the paper before publication.

While the paper does make a valuable scientific contribution with the derivation of equation (23), there are three things that would make the paper much stronger, and a more useful scientific contribution:

1. A more thorough analysis and discussion of the why the temperatures derived from the CPEC310 measurements in the field and equation (23) did not more closely match the temperature used as a reference (the platinum resistance thermometer (PRT) inside the fan-aspirated radiation shield). There is a small section about this in lines 440-445. The suggestion that the PRT inside the fan-aspirated shield might be reading low, in lines 442-443, could be further investigated. For example, was the PRT calibrated before or after it was used to make measurements in the field? If so, was it reading low? It is likely that the PRT was not reading low, at least not by about 0.5 C, which suggests the temperatures derived from equation (23) are biased high (about 0.55 C in January and about 0.44 C in July). In lines 517-518 there is some brief commentary on systematic error in Ts measurements due to fixed deviation in measurements of sonic path lengths. Seems possible that this systematic error is the source of the bias in temperature from equation (23) when compared to temperature from the PRT in the fan-aspirated shield. There is also a suggestion that some of the difference in temperature may be due to non-ideal weather conditions, in lines 497-502. Given that weather variables were measured, it should be possible to filter the data for ideal weather conditions.

2. A more thorough analysis and discussion of the error in temperature computed from equation (23) in relation to the applications. From lines 544-558, it seems the main application of computing temperature from equation (23) and high frequency measurements is an accurate estimate of high frequency dry air density ($\rho_d$) for water vapor flux calculations, and calculation of sensible heat flux from air temperature, without the need of a humidity correction. If this is the case, then it would be useful to have an indication of how accurate high frequency $\rho_d$ and air temperature need to be and some commentary on whether the temperatures derived from the field data collected in this study and equation (23) are within this accuracy range. As stated above, it appears from field data that fixed deviation in the sonic path length may be the cause of the bias of about 0.5 C. If high frequency air temperature is high by about 0.5 C when computed with equation (23), can it practicably be used to improve flux calculations?

3. This study was conducted with only Campbell Scientific instruments. It would be helpful if there was some commentary on use of the proposed technique with other instruments. While Campbell Scientific instruments are widely used for flux measurements using the eddy covariance technique, other companies make 3D sonic anemometers and high frequency gas analyzers that are also widely applied for eddy covariance. Even if brief, any discussion the authors can provide about applicability of the proposed technique with non-Campbell Scientific instruments will make the paper more general. Right now, the information in the paper is specific to only those users who have Campbell Scientific instruments.

Beyond these three content recommendations, there are two things that would improve organization of the paper:

1.  Move sections 6 and 7 to an appendix. These sections contain important material, but provide a level of detail that is not essential to the main body of the paper.
2.  Use headers to better separate the material. For example, section 1 is the Introduction, section 2 is the Background, and section 10 is the Discussion. Following this formant, sections 3 and 4 could be called Theory. Section 5 could be called Materials and Methods. If sections 6 and 7 are not moved to an appendix, they should be included with section 5 under Materials and Methods. Sections 8 and 9 could be called Results.

Some necessary edits:

Line 17: temperar should be temperature.

Line 20: senosrs should be sensors.

Line 30: CPEC300 is a specific product and needs to be defined (meaning the instruments included with this model should be listed and the manufacturer should be listed).

Line 44: Panofsky and Dotton (1984) is cited, but is not found in the reference list.

Line 66: suffered to should be changed to suffered by.

Line 110: contaminated should be contamination.

Line 152: a should be removed after unmeasurable by.

Lines 243-244: CSAT3A and EC155 are specific Campbell Scientific products, so they should be denoted as such (like at the beginning of the sentence where CPEC310 is denoted as a Campbell Scientific product).

Line 262: Multiple temperature variables are used in equation (27). Subscript c appears to denote calibration, subscript z appears to denote zero, subscript s appears to denote sonic, and unclear what subscript r denotes. Some clarification and definition is required.

Lines 323-324: EC100 is a specific Campbell Scientific product and needs to be denoted as such. It seems the EC100, EC155, and CSAT3A are all components of the CPEC310. If this is the case, it would be helpful if there is a better description of the CPEC310.

Line 341: CR6 needs to be defined as a datalogger and denoted as a Campbell Scientific product.

Line 357: Sentence needs to be reworded. The phrase even impossible is out of place. Perhaps remove the phrase even impossible from the sentence and then write another sentence to describe how it is impossible to sample fast enough to capture all eddies.

Line 527: Acronym OPEC is used without being defined. Needs to be defined as open path eddy covariance.

Line 609: thermometry can be removed.

Line 674: Diving should be dividing by.

Lines 682 and 685: expending should be expanding.

---

## Author Comment (AC1)

**Response to the Comments from Referee #1 on "Air temperature equation derived from**
**sonic temperature and water vapor mixing ratio for air flow sampled through closed-path**
**eddy-covariance flux systems"**

X.H. Zhou, T. Gao, E.S. Takle, X.J. Zhen, A.E Suyker, T. Awada, J. Okalebo, J.J. Zhu
https://doi.org/10.5194/amt-2021-160

The sentences in bold font are our responses to the underlined comments.

Air temperature is certainly a very important parameter for describing the state of the
atmosphere from high-frequency turbulence to climatological means. There are very reliable
and inexpensive measuring instruments for this purpose. It certainly makes sense to look for
a measuring method that can accurately measure the air temperature without the influence
of solar radiation (radiation error).

**The purpose of the paper is not specifically to eliminate solar radiation contamination – it is**
**to find an exact equation of air temperature in terms of sonic temperature and water vapor**
**mixing ratio and to develop the methodologies from this equation for better measuring**
**"turbulent $T$" for combining with concurrently measured turbulent 3D wind speeds to**
**represent turbulent heat flux and related turbulent variables. The insensitivity of derived $T$**
**to solar radiation is an expected additional merit of the derived $T$.** For this purpose,
ventilated thermometer screens are used for very accurate measurements. This is a good way
to meet the World Meteorological Organisation's requirement of an accuracy of $\pm$ 0.2 K at
0 °C (WMO, 2018). **Our senior authors have worked on turbulence measurements over 30**
**years. To the best of our knowledge, we have not been aware that WMO has a standard for**
**"high-frequency $T$". WMO (2018) requirements are for common low-frequency $T$ of weather**
**and climate network stations instead of "high-frequency $T$" in turbulent flux measurements.**
**The Fine Wire Thermocouples (i.e. FW series. see https://www.campbellsci.com/fw05),**
**which are most commonly used for the high-frequency $T$ measurements in flux community,**
**do not have specifications for accuracy and precision. Authors have used such an option over**
**20 years and programmatically implemented such measurements into EasyFlux series**
**software for global use as optional measurements for users. However, this option cannot be**
**used for long-term measurements because it is fragile as discussed in the manuscript. The**
**experts on manufacturing FW sensors were consulted about unavailability for the**
**specifications of accuracy and precision. Simple answers are a) the method to specify**
**accuracy and precision for high-frequency $T$ is not available and b) no standard for high-**
**frequency $T$ can be followed. Accordingly, this comment is not relevant to this study.** Even
with naturally ventilated thermometer screens, this accuracy can be achieved in many cases
(Harrison and Burt, 2021). **The reviewer misses the major point (Lines 60 to 65 in**
**Introduction and lines 89 to 104 in Background) that accurate measurement of turbulent**
**heat fluxes and related variables, which is the goal of the paper, cannot be done with a**
**ventilated thermometer co-located with a sonic wind speed measurement. Furthermore,**

ventilated thermometers report only time-averaged (rather than at turbulence frequencies)
temperature values.

It therefore seems somewhat absurd - if I have understood the authors correctly - to use
device combinations of sonic anemometers and closed-path gas analysers to obtain an
accurate temperature measurement, especially since operators of these systems often also use
a simple temperature-humidity sensor for quality assurance. **Simple temperature-humidity**
**measurements can provide quality assurance of mean, but not turbulent, $T$.** This request of
the authors seems all the more doubtful, as the requirement of measuring accuracy for
temperature measurements is not achieved. However, an accuracy of ± 1 K is quite sufficient
to determine the temperature-dependent densities and specific heats for trace gas
measurements. **But ±1K is not acceptable for turbulent fluxes at high frequency.** In most
cases, the sonic temperature can be used directly, if necessary with a small correction. **Direct**
**use of sonic temperature as $T$ has a great uncertainty under warm and humid conditions and**
**is not an acceptable approximation. What the sonic anemometer reports is sonic**
**temperature, which requires knowledge of air humidity to calculate the actual air**
**temperature (i.e., $T$). Actual air temperature and sonic temperature are quite different.**
**Given $T$ = 35 °C and RH = 100%, the difference is 5.6 °C. Under the same humidity, given $T$ =**
**45 °C, this difference reaches 10 °C.** The authors start from the basic work on the conversion
of sonic temperatures into air temperatures (Kaimal and Gaynor, 1991; Schotanus et al.,
1983). At first sight, the calculation seems to be correct. However, due to the deviousness of
the procedure, no examination in detail was carried out. **The reviewers should explain what**
**they mean by the "deviousness of the procedure", which suggests a dishonest intent. The**
**calculation is a bit complex, but in no way is it dishonest. Perhaps this was simply a poor**
**choice of words. If so, the reviewers should find a more specific word so that more clarity of**
**the procedure can be provided.**

The authors used a sonic anemometer, which allows a fairly accurate measurement of the
sonic temperature. Since the measurement depends strongly on the mechanical stability of
the device, there are also devices with much worse values (Mauder and Zeeman, 2018) with
deviations up to several kelvin, so that the proposed method is only applicable for selected
types of sonic anemometers. **Of course, there may be some sonic instruments that have large**
**errors in sonic temperature measurement. The authors test only one state-of-the-art sonic**
**instrument which is designed with both hardware configuration and instrument-specific**
**software developed from turbulence theory based on fundamental principles. Their reported**
**detailed tests show high accuracy can be achieved for measuring $T$ at high frequency. Due to**
**different grade of accuracies from different brands of sonic anemometers (e.g., CAST, Gill,**
**and Young), one of our major objectives is to avoid the direct error of turbulent $T$ from**
**theoretical equation side. The indirect error from sonic anemometers for sonic temperature**
**and from gas analyzer for air moisture goes beyond the scope of this paper. For the objectives**
**of this study, there is no need to test this equation by more sonic instruments in the field.**

The reviewer strongly doubts that there is a reader of AMT who would find this method
interesting for application. This doubt is subjective instead of objective. The following three
points disagree with reviewer's doubt.

**1.**      This study has been driven by applications of sonic temperature and water vapor
mixing ratio for sensible heat flux. When the first author started his EasyFlux_CR6CP for
close-path eddy-covariance (CPEC) systems in Campbell Scientific Inc., he needed an
equation for sensible heat flux from sonic temperature and water vapor mixing ratio.
Definitely, Schotanus et al. (1983), Kaimal and Gaynor (1991), and van Dijk (2002) were
under consideration, but the problem was found as addressed in Introduction and
Background. We thoroughly studied the relationship of sonic temperature and air moisture
to $T$ and derived the exact equation of $T$ in terms of sonic temperature and water vapor
mixing ratio. For field applications of this equation to CPEC systems, we also developed
algorithms as addressed in the manuscript. As well known, Schotanus et al. (1983), Kaimal
and Gaynor (1991), and van Dijk (2002), all of which did not have uncertainty specifications
and field tests for high-frequency $T$, have wide applications in flux community. Our exact
equation avoids the uncertainty/controversies from their equations with additional field
tests. It is better developed, tested, and documented as in the manuscript.

As always, exact equations are pursued tirelessly by scientists, so replacing the
approximate equations with an exact equation represents a scientific advance for field
measurement (this assertion is validated by reviewer #2). Now, after verification against
sensible heat flux measurements from a fine wire thermocouple configured in a CPEC
system, this equation has been used in the open-source software EasyFlux-DL-CR6CP
([https://www.campbellsci.com/revisions/626-1506#revisions](https://www.campbellsci.com/revisions/626-1506#revisions)). This software is being used
globally for hundreds of Campbell Scientific CPEC systems deployed in the field (e.g., 30 in
New York Mesonet and 36 new orders to China). China alone has over 100 CPEC systems in
the field. CPEC systems are recommended systems, due to better data continuity and
reliability, now as demonstrated in a China national field laboratory (Zhu et al. 2021). For
their customized use of EasyFlux-DL-CR6CP, the users of hundreds of field CPEC systems
deserve to fully understand the equations used in the software from a formal journal like
AMT. If the reviewer can show where the theory and derivation of our paper is invalid,
he/she should point out the flaw so that the field implementation of the algorithm is changed
for more accurate measurements.

**2.**      Manufacturers of sonic anemometers need the exact equation, instead of
approximation ones, to improve the manufacturing process. For precision measurements of
sonic temperature along with 3D wind, the lengths of three sonic anemometer paths are
precisely measured physically by Coordinate Measurement Machine (CMM) in
manufacturing process. CMM has some limitations in the length measurements to achieve
the accuracy of sonic temperature to high accuracy (e.g. < ±1.00 K). From the measurements
of $T$ and water vapor mixing ratio, sonic temperature can be accurately determined if an exact equation to describe the relationship among the three variables is available. Using this
accurate sonic temperature, the sonic path lengths can be theoretically acquired better than
CMM. See Zhou et al (2018) for the relationship of sonic temperature to the path lengths.
This technology is under development. The exact equation, which has been pursued since
1932 (Ishii 1932, Barrett and Suomi 1949, Schotanus et al. 1983, Kaimal and Gaynor 1991,
Swiatek 2018), is fundamental, prerequisite, and valuable, in particular, for this technology.
For commercial rules, it is inappropriate for authors to disclose more details of this
technology here. Our brief disclosure can say the exact equation is "exactly valuable" for the
advancement of sciences and technology, which is a common sense in scientific community.

3.      From June 21 this year until now (less than two months), as recorded by AMT
editorial website, this manuscript has received 273 views, and 58 XML and PDF download
actions from the limited number of viewers of five countries. These metrics provided by
AMT indicate the interest of this manuscript to AMT readers. If formally published, more
readers can access this information. This topic would be of interest to AMT readers in the
same way as this topic is often asked by the audience in international training courses (e.g.
Annual ChinaFlux training courses) by the first author.

References
Barrett, E. W. and Suomi, V. E.: Preliminary report on temperature measurement by sonic
means, J. Atmos. Sci., 6, 273–276, 1949.

Harrison, R. G., and Burt, S. D.: Quantifying uncertainties in climate data: measurement
limitations of naturally ventilated thermometer screens, Environ. Res. Commun., 3,
1-10, doi: 10.1088/2515-7620/ac0d0b, 2021.

Ishii, C.: Supersonic velocity in gases: especially in dry and humid air. Scientific Papers of the
Institute of Physical and Chemical Research, Tokyo, 26, 201–207 pp., 1932.

Kaimal, J. C., and Gaynor, J. E.: Another look to sonic thermometry, Boundary-Layer
Meteorol., 56, 401-410, 1991.

Mauder, M., and Zeeman, M. J.: Field intercomparison of prevailing sonic anemometers,
Atmos. Meas. Tech., 11, 249-263, doi: 10.5194/amt-11-249-2018, 2018.

Schotanus, P., Nieuwstadt, F. T. M., and DeBruin, H. A. R.: Temperature measurement with
a sonic anemometer and its application to heat and moisture fluctuations, Boundary-
Layer Meteorol., 26, 81-93, 1983.

Swiatek, E: Derivation of Temperature (Tc) from the Sonic Virtual Temperature (Ts), vapor
density ($\rho v$)/vapor pressure (e) and pressure (P). Campbell Scientific Inc. Logan, UT,
1-5 pp., 2018

WMO: Guide to Instruments and Methods of Observation, WMO-No. 8, Volume I -
Measurement of Meteorological Variables, World Meteorological Organization,
Geneva, 548 pp., 2018.

van Dijk, A.: The principles of surface flux physics. Department of Meteorology and Air
Quality, Agriculture University Wageningen, 40–41 pp., 2002.

Zhou, X., Yang, Q., Zhen, X., Li, Y., Hao, G., Shen, H., Gao, T., Sun, Y., and Zheng, N.:
Recovery of the three-dimensional wind and sonic temperature data from a physically
deformed sonic anemometer, Atmos. Meas. Tech., 11, 5981–6002, doi:10.5194/amt-11-
5981-2018, 2018.

Zhu, J.J., Gao, T., Yu, L.Z., Yu, F.Y., Yang, K., Lu, D.L., Yan, Q.L., Sun, Y.R., Liu, L.F., Xu, S.,
Zhang, J.X., Zheng, X., Song, L.N., Zhou, X.H. Functions and applications of Multi-
tower Platform of Qingyuan Forest Ecosystem Research Station of Chinese Academy
of Sciences (Qingyuan Ker Towers). Bulletin of Chinese Academy of Sciences 3, 351-
361, 2021.

---

## Author Response (AR1)

**INSTITUTE OF APPLIED ECOLOGY, CHINESE ACADEMY OF SCIENCES**

**72 Wenhua Road, Shenyang, Liaoning, 110016, China**

Oct 14, 2021

RE: Revision for amt-2021-160

Dr. Keding Lu
College of Environmental Science and Engineering
Peking University
Beijing 100871, China

Dear Dr. Lu,

First of all, we thank you so much for your allowing us to revise our manuscript "Air temperature equation derived from sonic temperature and water vapor mixing ratio for turbulent air flow through closed-path eddy-covariance flux systems" for further consideration in *Atmospheric Measurement Techniques* (AMT).  As we addressed in our final author comments/response in Sept 13, 2021 and more discussions with our co-authors, we have completed our revision. After revision, authors checked the manuscript. Additionally, Ms. Brittney Smart professionally proofread the manuscript throughout. Her edits were incorporated into this revision. The revision was proofread by Dr. Xinhua Zhou again while checking the consistency throughout for expressions.

The line and section numbers in our response to referees' comments and in author revisions are referred to those in previous version: https://doi.org/10.5194/amt-2021-160 because both numbers are changing in revision process.

We appreciate your further consideration for our manuscript publication in AMT.

Sincerely,

Tian Gao, Ph.D., Research Associate Professor
Remote Sensing for Forest Fluxes and Management

**Response to Referees' comments on "Air temperature equation derived from sonic temperature and water vapor mixing ratio for air flow sampled through closed-path eddy-covariance flux systems"**

X.H. Zhou, T. Gao, E.S. Takle, X.J. Zhen, A.E Suyker, T. Awada, J. Okalebo, J.J. Zhu

**Response to Referee #1**

We thank Referee #1 very much for his/her discussion comments although without evaluation on either equation derivation merits or technical flaws. The discussion comments are valuable for us to think deeper and improve the manuscript in revision process.

Referee #1 did not number his/her discussion comments. For clarity, the comments are separated into nine. The underlined sentences in each numbered comment are responded in blue.

1. Air temperature is certainly a very important parameter for describing the state of the atmosphere from high-frequency turbulence to climatological means. There are very reliable and inexpensive measuring instruments for this purpose. It certainly makes sense to look for a measuring method that can accurately measure the air temperature without the influence of solar radiation (radiation error).

   *Author response*
   The purpose of the paper is not specifically to eliminate solar radiation contamination – it is to find the exact equation of air temperature ($T$) in terms of sonic temperature ($T_s$) and water vapor molar mixing ratio ($\chi_{H_2O}$) and to develop the methodologies from this equation for better measuring "turbulent $T$" for combining with concurrently measured turbulent 3D wind speeds to represent turbulent heat flux and related turbulent variables. The insensitivity of equation-computed $T$ to solar radiation is an expected additional merit.

   *Author revision*
   In the end of line 508 in Section 9.2, one more sentence is added: "*Although the purpose of this study is not particularly to eliminate solar radiation contamination to turbulent T, equation-computed T is indeed less contaminated by solar radiation as shown in Fig 6.*"

2. For this purpose, ventilated thermometer screens are used for very accurate measurements. This is a good way to meet the World Meteorological Organisation's requirement of an accuracy of $\pm$ 0.2 K at 0 °C (WMO, 2018).

   *Author response*
   Our senior authors have worked on turbulence measurements over 30 years. To the best of our knowledge, we have not been aware that WMO has a standard for "high-frequency $T$". WMO (2018) requirements are for common "low-frequency $T$" of weather and climate network stations instead of "high-frequency $T$" in turbulent flux measurement. Fine wire thermocouples (i.e., FW series. https://www.campbellsci.com/fw05), which are most commonly used for the high-frequency $T$ measurement in flux community, do not have specifications for accuracy and precision. Authors have used such an option over 20 years and programmatically implemented such a measurement into EasyFlux series software for global use as an optional measurement by users. However, this option cannot be used for long-term measurements because FW sensors are fragile as discussed in the manuscript (see lines 105 and 106). The experts, Senior Engineers: Edward Swiatek and Antoine Rousseau,

on manufacturing FW sensors were consulted about unavailability for the specifications of accuracy and precision. Simple answers are a) the method to specify accuracy and precision for high-frequency $T$ is not available and b) no standard for high-frequency $T$ can be followed. Accordingly, this comment is not relevant to this study on high-frequency turbulent $T$.

*Author revision*
Throughout manuscript in revision, the authors emphasize the natural of equation-computed T in frequency response. In N/A. Also see our response in 5.

3. Even with naturally ventilated thermometer screens, this accuracy can be achieved in many cases (Harrison and Burt, 2021).

*Author response*
The referee misses the major point (Lines 60 to 65 in the Introduction and lines 89 to 104 in the Background) that the accurate measurement of turbulent heat fluxes and related variables, which is the goal of the paper, cannot be done with a ventilated thermometer co-located with a sonic wind measurement. Furthermore, ventilated thermometers report only time-averaged (rather than at turbulence fluctuations at high frequencies, e.g., 10 Hz) $T$ values. In the Background, the authors re-emphasize and clarify that high-frequency $T$ cannot be acquired from a ventilated (i.e., aspirated) thermometer.

*Author revision*
a.  The sentence in lines 105 and 106 was revised as "*Additionally, aspiration methods cannot acquire T at high frequency due to the disturbance of an aspiration fan to natural turbulent flows and fine wire thermocouples have limited applicability for long-term measurements in rugged field conditions typically encountered in ecosystem monitoring.*

b.  In title, "air flow" was changed into "*turbulent air flow*".

4.  It therefore seems somewhat absurd - if I have understood the authors correctly - to use device combinations of sonic anemometers and closed-path gas analysers to obtain an accurate temperature measurement, especially since operators of these systems often also use a simple temperature-humidity sensor for quality assurance.

*Author response*
Simple $T$ and RH (relative humidity) measurements can provide quality assurance of mean, but not turbulent $T$. This is discussed while variables for Eqs. (2) and (3) are discussed.

*Author revision*
N/A

5.  This request of the authors seems all the more doubtful, as the requirement of measuring accuracy for temperature measurements is not achieved. However, an accuracy of $\pm$ 1 K is quite sufficient to determine the temperature-dependent densities and specific heats for trace gas measurements.

*Author response*
The accuracy of $\pm 1.0$ K for high frequency-$T$ is more acceptable in turbulent flux measurement because it is the best accuracy of $T_s$ from individual sonic anemometers required for sensible heat flux in all CPEC (closed-path eddy-covariance) systems.

*Author revision*
In section 4.3.3, one sentence as the last is added. "*This accuracy for high frequency-T currently is the best in turbulent flux measurement because ±1.0 K is the best in accuracy of $T_s$ from individual sonic anemometers required for sensible heat flux in all CPEC systems*".

6.  In most cases, the sonic temperature can be used directly, if necessary with a small correction.

*Author response*
Direct use of $T_s$ as $T$ has a great uncertainty under warm and humid conditions and is not an acceptable approximation (Kaimal and Gaynor 1991, Schotanus et al. 1983). What the sonic anemometer reports is $T_s$, which requires knowledge of air humidity to calculate the actual air temperature (i.e., $T$).  Actual $T$ and $T_s$ are quite different. Given $T = 35$ °C and RH = 100%, the difference is 5.6 °C. Under the same RH, given $T = 45$ °C, this difference reaches 10 °C.

*Author revision*
N/A

7.  The authors start from the basic work on the conversion of sonic temperatures into air temperatures (Kaimal and Gaynor, 1991; Schotanus et al., 1983). At first sight, the calculation seems to be correct. However, due to the deviousness of the procedure, no examination in detail was carried out.

*Author response*
The referee should explain what they mean by the "deviousness of the procedure", which suggests a dishonest intent. The calculation is a bit complex, but in no way is it dishonest. Perhaps this was simply a poor choice of words. If so, the referee should find a more specific word so that more clarity of the procedure can be provided.

*Author revision*
N/A

8.  The authors used a sonic anemometer, which allows a fairly accurate measurement of the sonic temperature. Since the measurement depends strongly on the mechanical stability of the device, there are also devices with much worse values (Mauder and Zeeman, 2018) with deviations up to several kelvin, so that the proposed method is only applicable for selected types of sonic anemometers.

*Author response*
Of course, there may be some sonic instruments that have large errors in $T_s$ measurement. The authors test only one state-of-the-art sonic instrument which is designed with both hardware configuration and instrument-specific software developed from turbulence theory based on fundamental principles. Their reported detailed tests show that a high accuracy can be achieved for high-frequency $T$ (Figs. 4 and 6). Due to different grades of accuracies from different models and brands of sonic anemometers (e.g., CAST3, Gill, and R.M. Young), one of our major objectives is to avoid the direct error of turbulent $T$ from theoretical equation side.  The indirect error from sonic anemometers for $T_s$ and from infrared analyzer for $\chi_{H2O}$ is unavoidable. Although this indirect error is becoming smaller with the

improvement of manufacturing technologies [e.g., CSAT3A ($\pm$2.0 °C) and updated CSAT3A ($\pm$1.0 °C)], the reduction in this error goes beyond the scope of this study.

However, the theory to evaluate the accuracy of equation-computed $T$ from any hardware combination of sonic anemometers and infrared analyzers is still in the scope of this study. As such, error equations (Eqs 24-27) are developed. Given hardware specifications, the accuracy of equation-computed $T$ from any hardware combination can be estimated by the error equations as shown in Fig. 2. Apparently, our theory on $T$ equation in terms of $T_s$ and $\chi_{H2O}$ is fully developed, which has not been done by any of previous studies (Ishii 1932, Barrett and Suomi 1949, Schotanus et al. 1983, Kaimal and Gaynor 1991, Swiatek 2018). Our theory is applicable to all combinations of sonic anemometers and infrared analyzers. For the objectives of this study, there is no need to test this equation through more combinations of sonic and infrared instruments in the field. Acknowledging this concern by Referee #1, we could add more explanations to section 3.5.

*Author revision*

Inserted four sentences into line 237 after "…… 2018a).", saying: "*Sonic anemometers and infrared analyzers with different models and brands have different specifications from their manufacturers. Any combination of sonic and infrared instruments has a combination of the $\Delta T_s$ and $\Delta \chi_{H2O}$ that are specified by their manufacturers. In turn, from Eq. (25), the combination generates $\Delta T$ of equation-computed $T$ for the corresponding combination of the sonic and infrared instruments with given models and brands. Therefore, Eqs. (23) and (25) are applicable to any CPEC system beyond our study brand.*"

9. The reviewer strongly doubts that there is a reader of AMT who would find this method interesting for application.

*Author response*
This doubt is subjective instead of objective. The following three facts disagree with referee's doubt.

**a.** This study has been driven by applications of $T_s$ and $\chi_{H2O}$ for sensible heat flux ($H$). When the first author started his EasyFlux-DL-CR6CP for CPEC systems in Campbell Scientific Inc., he needed an equation for $H$ from $T_s$ and $\chi_{H2O}$. Definitely, Schotanus et al. (1983), Kaimal and Gaynor (1991), and van Dijk (2002) were under consideration, but the disparity among available equations was found as addressed in the Introduction and the Background. We thoroughly studied the relationship of $T_s$ and air moisture to $T$ and derived the exact equation of $T$ in terms of $T_s$ and $\chi_{H2O}$. For the applications of this equation to field CPEC systems, we also developed algorithms as addressed in section 7. As well known, Schotanus et al. (1983), Kaimal and Gaynor (1991), and van Dijk (2002), all of which do not have uncertainty specifications and field tests for high-frequency $T$, have wide applications in flux community. Our exact equation prevents the uncertainty/controversies from their equations also with additional field tests. It is better developed, tested, and documented as in this manuscript.

As always, an exact equation is pursued tirelessly by scientists, so replacing an approximate equation with the exact one represents a scientific advance for field measurement (this assertion is validated by Referee #2 for this version and two referees for the previous version). Now, after verification in development tests against $H$ measurements from a fine

wire thermocouple configured in a CPEC system, this equation has been used in the open-source software EasyFlux-DL-CR6CP (https://www.campbellsci.com/revisions/626-1506#revisions). This software is being used globally for hundreds of Campbell Scientific CPEC systems deployed in the field (e.g., 30 in New York Mesonet and almost 100 new orders to China). China alone now has over 100 CPEC systems mostly in the northwest region (e.g., Tibet and Gansu regions). CPEC systems are recommended systems, due to better data continuity and reliability, now as demonstrated in a China national field laboratory (Zhu et al. 2021). For their customized use of EasyFlux-DL-CR6CP, the users of hundreds of field CPEC systems deserve to fully understand the equations used inside the software from a formal journal like AMT. If the referee can show where the theory and derivation of our paper is invalid, he/she should point out the flaw so that the field implementation of the algorithm is changed for more accurate measurements. Current applications are addressed in this revision.

**b.** This exact $T$ equation has wide applications. The scope of this study is to derive the exact $T$ equation (section 3), verify the accuracy of equation-computed $T$ (Figs. 2 and 4), test frequency response (Fig. 5) and other merit (Fig. 6), and brief the perspectives for applications (section 10.3). The deeper discussion on equation applications goes beyond the scope. Because of length of manuscript, more analyses of applications go beyond the scope even further. Full contents in the previous version were the major reason for Dr. Massman to recommend this manuscript to be resubmitted. For referee's concern, in addition to section 10.3, we would give an additional application example.

Manufacturers of sonic anemometers need the exact equation, instead of approximate one, to improve the manufacturing process. For precision measurement of $T_s$ along with 3D wind, the lengths of three sonic anemometer paths are precisely measured physically by the Coordinate Measurement Machine (CMM) in manufacturing process. So far, the CMM has some limitations in the length measurements to achieve the accuracy for $T_s$ to be significantly better than $\pm 1.0$ K. $T$ and $\chi_{H2O}$ can be more accurately measured under manufacturing environment. Using the accurate measurements, $T_s$ can be accurately determined by Eq. (20) equivalent to Eq. (23). Using this accurate $T_s$, the sonic path lengths can be theoretically acquired better than physically measured from the CMM. See Zhou et al (2018) for the relationship of $T_s$ to the path lengths. This technology is under development. The exact equation, which has been pursued since 1932 (Ishii 1932, Barrett and Suomi 1949, Schotanus et al. 1983, Kaimal and Gaynor 1991, Swiatek 2018), is fundamental, prerequisite, and valuable, in particular, for this technology. For commercial rules, it is inappropriate for authors to disclose more details of this technology here. Our brief disclosure between authors and referees can reveal the exact equation is "exactly valuable" for the advancement of sciences and technologies, which is a common sense in scientific community. The discussion in this paragraph may be inappropriate to be added to the manuscript.

**c.** From June 21 this year until now, as recorded by AMT editorial website, this manuscript has received 400 reviews, and 107 XML and PDF download actions from the limited number of public reviewers all over the world (Fig. 1). These metrics provided by AMT indicate the interest of this manuscript to AMT readers. If formally published, more readers can access this information. This topic would be of interest to AMT readers in the same way

as this topic is often asked by the audience in international training courses (e.g., Annual ChinaFlux training courses) given by the first author.

*Author revision*
N/A.

[Figure]

Figure 1. Matric for registered AMT audience to view and down load (Source: https://amt.copernicus.org/preprints/amt-2021-160/#discussion)

*Author revision*
a. The paragraph between lines 543 and 546 was replaced with

 *"The air temperature equation (23) is derived from first principles without any assumption and approximation. It is an exact equation from which T can be computed in CPEC systems as a high-frequency signal insensitive to solar radiation. These merits in additional to its consistent representation of spatial measurement and temporal synchronization scales with other thermodynamic variables for boundary-layer turbulent flows will be more needed for advanced applications. Popularly used in the world, EasyFlux series is one of the two field eddy-covariance flux software packages, in which the other included is EddyPro (LI-COR Biosciences, 2015). Currently, it has used equation-computed T for $\rho_d$ in Eq. (1), sensible heat flux (H), and RH as a high-frequency signal in CPEC systems (Campbell Scientific Inc. 2018a)."*

b. Below line 777, added:
*"LI-COR Biosciences: EddyPro® eddy covariance software: instruction manual, Lincoln, NE, USA, 1—1 to 10—6 pp., 2015."*

**Response to Referee #2**

**Overall Comment**

The paper is clearly written and makes a valuable scientific contribution with the derivation of equation (23), an equation based on first principles that can be used to compute high frequency air temperature from measurements with closed path eddy covariance systems. As shown in the paper, equation (23) is an advance beyond the previously used approximations, equations (4) and (5). And, as described in the paper, high frequency air temperature computed from equation (23) has potential to improve flux calculations with the eddy covariance technique. The paper should be published, but full consideration should be given the comments below, as there is opportunity to improve the paper before publication. While the paper does make a valuable scientific contribution with the derivation of equation (23), there are three things that would make the paper much stronger, and a more useful scientific contribution:

> *Author response*
> We thank Referee #2 so much for his/her professional review, understanding on our study topic, and constructive comments on improvement of manuscript. This overall comment agrees with the overall comments from the two referees for the previous version.

**Discussion Comments**

1. A more thorough analysis and discussion of the why the temperatures derived from the CPEC310 measurements in the field and equation (23) did not more closely match the temperature used as a reference (the platinum resistance thermometer (PRT) inside the fan aspirated radiation shield). There is a small section about this in lines 440-445. The suggestion that the PRT inside the fan-aspirated shield might be reading low, in lines 442-443, could be further investigated. For example, was the PRT calibrated before or after it was used to make measurements in the field? If so, was it reading low? It is likely that the PRT was not reading low, at least not by about 0.5 C, which suggests the temperatures derived from equation (23) are biased high (about 0.55 C in January and about 0.44 C in July). In lines 517-518 there is some brief commentary on systematic error in Ts measurements due to fixed deviation in measurements of sonic path lengths. Seems possible that this systematic error is the source of the bias in temperature from equation (23) when compared to temperature from the PRT in the fan-aspirated shield. There is also a suggestion that some of the difference in temperature may be due to non-ideal weather conditions, in lines 497-502. Given that weather variables were measured, it should be possible to filter the data for ideal weather conditions.

> *Author response*:
> This comment is related to several issues. We categorize the issues into four: a. PRT calibration, b. PRT (platinum resistance thermometer) accuracy, c. Equation error, and d. Data filtering. We address the issues separately in four categories as followings:
>
> a. PRT calibration
> Both CSAT3A (updated version) and PRT sensor were new when the field tests for this study were started, but both were not recalibrated after the field tests because both sensors continued to be employed for ongoing program tests. The data from the field as shown in Fig. 4 verified the relationship of the equation-computed $T$ (air temperature) error to sonic anemometer and infrared analyzer specifications [i.e., measurement errors in sonic temperature ($T_s$) and water vapor mixing ratio ($\chi_{H2O}$). See Eq. 25 and Figs. 2 and 4]. Although

PRT may drift in $T$ measurement and CSAT3A may drift in $T_s$ measurement, for convincing analysis, we better use the specified accuracies of $T$ from PRT ($\pm0.2$ °C) and $T_s$ from CSAT3A ($\pm1.0$ °C). Although equation-computed $T$ is biased high about 0.55 C in January and about 0.44 C in July, as shown in Fig. 2, the biases are within the range as described by Eq. (25) in terms of sensor specifications. We would be criticized by other referees if we further narrow the accuracy range of equation-computed $T$ through finding a greater error from either $T_s$ or PRT-measured $T$.

b. PRT accuracy
See line 443, $\pm0.2$ instead of $\pm0.5$ °C is used for discussion on PRT accuracy. It is well-known, the PRT accuracy in $T$ is specified as $\pm0.2$ °C by R.M. Young. The referee wrongly read the accuracy in Line 443.

c. Equation error
We believe that our developed $T$ equation (Eq. 23) does not have any error. The equation-computed $T$ has an error, but the error arises from the measurement instead of equation. The measurement errors are in $T_s$ and $\chi_{H2O}$. The error of equation-computed $T$ was analytically expressed in terms of $T_s$, $\chi_{H2O}$, and their measurement accuracies in Eq. (25) (i.e., error equation). The $T$ equation and its error equation are exact ones. The derivation of both equations has been reviewed by four referees and checked by our co-authors.

It is common for sonic anemometers to have a systematic error in $T_s$ to be $\pm0.5$ ºC or little greater, which is the reason that the $T_s$ accuracy is specified by Larry Jacobsen (anemometer authority) to be $\pm1.0$ ºC. The fixed deviation in measurements of sonic path lengths is asserted as the source for bias of $T_s$ (Zhou et al., 2018). This bias brings an error to equation-computed $T$. If the $T$ equation were not exact, the error in equation-computed $T$ would be introduced not only from $T_s$ and a $\chi_{H2O}$ errors, but also from an equation error. The objective of this study is to avoid equation error. Referee #2 categorizes the measurement error in $T_s$ into the equation error. The reduction in a measurement error relies on the manufacturing technologies. In turn, the application of this exact equation can be used to improve this technology (see our response b to Referee #1's comment 9).

Acknowledging this point of Referee #2, we may need further clarifying discussion among lines 440 to 445.

d. Data filtering
The data for this study were not subjectively filtered. The data of the sonic anemometer and infrared analyzer were filtered by their measurement diagnosis (see lines 549-551), but the data from PRT were not filtered. Subjectively filtering data would bring controversy to data analyses. The quality of data shown in Fig. 4 are the real quality of field data from the close-path eddy-covariance (CPEC) system.

We suspect that unfavorable weather contributed to a $T_s$ error. We could have filtered out unfavorable weather cases to create a lower error estimate. But since most field experiments include periods when weather increases a $T_s$ error, we reasoned that including a weather contribution to error would avoid overstating instrument accuracy under typical (unfiltered) applications.

Acknowledging this point of Referee #2, we need more clarification of discussion in section 10.1.

*Author revision*

a. Revision is not needed.

b. Revision is not needed.

c. The last sentence in line 444 is replaced with the following paragraph:

"*It is common for sonic anemometers to have a systematic error in $T_s$ to be $\pm 0.5\ ^oC$ or little greater, which is the reason that the $T_s$ accuracy is specified by Larry Jacobsen (anemometer authority) to be $\pm 1.0\ ^oC$ for updated CSAT3A. The fixed deviation in measurements of sonic path lengths is asserted as a source for bias of $T_s$ (Zhou et al., 2018). This bias brings an error to equation-computed T. If the T equation were not exact as in Eqs. (4) and (5), there would be an additional equation error. In our study effort, this bias from fixed deviation possibly is around 0.5 $^oC$. With this bias, the equation-computed T is still accurate as specified by Eqs. (25) to (27) and even better.*"

d. Add a short paragraph in the end of section 10.1.

"*For this study, filtering out the $T_s$ data in the periods of unfavorable weather could narrow the error range of equation-computed T. The unfavorable weather was suspected to contribute the stated error. However, although filtering out unfavorable weather cases could create a lower error estimate, most field experiments include periods when weather increases a $T_s$ error, so including a weather contribution to error would prevent overstating instrument accuracy under typical (unfiltered) applications. Therefore, both $T_s$ and $\chi_{H2O}$ data in this study were not programmatically or manually filtered based on weather.*"

2. A more thorough analysis and discussion of the error in temperature computed from equation (23) in relation to the applications. From lines 544-558, it seems the main application of computing temperature from equation (23) and high frequency measurements is an accurate estimate of high frequency dry air density ($\rho_d$) for water vapor flux calculations, and calculation of sensible heat flux from air temperature, without the need of a humidity correction. If this is the case, then it would be useful to have an indication of how accurate high frequency $\rho_d$ and air temperature need to be and some commentary on whether the temperatures derived from the field data collected in this study and equation (23) are within this accuracy range. As stated above, it appears from field data that fixed deviation in the sonic path length may be the cause of the bias of about 0.5 C. If high frequency air temperature is high by about 0.5 C when computed with equation (23), can it practicably be used to improve flux calculations?

*Author response*

As our response to Referee #1, the deeper discussion on equation applications goes beyond the scope of this study (see response b to Referee #1's comment 9). For Referee's concern, in addition to section 10.3.2, additional discussion could be given below.

We can use the equation of state to estimate what the change in dry air density ($\rho_d$) can be caused from the difference in $T$ of $\pm 0.5$ °C. However, this estimation cannot be used to assess the practical improvement for flux calculations because, to assess the improvement, comparison is needed. This concern can be clarified by the following two explanations:

a. Currently, beyond Campbell Scientific flux software, Eqs. (4) and (5) are used for sensible heat flux computations. Both equations are all approximate equations (see Appendices A and B). Our equation is an exact one. Compared to either approximate equation, our exact equation must be an improvement on the mathematical representation of

sensible heat flux. If the equation for sensible heat flux is approximate, then even a perfect measurement gives only an approximate value for the flux. The explanations in this paragraph could be expected by Referee #2 to support the discussion among lines 544-558.

b. Currently, in $CO_2$, $H_2O$ and trace gas flux measurements, mean $\rho_d$ for flux calculations is estimated from $T$ and RH (relative humidity) along with atmospheric pressure. $T$ and RH are measured mostly by a slow-response $T$-RH probe without fan-aspiration (e.g., HMP155A, Vaisala Corporation, Helsinki, Finland) (Zhu et al. 2021). As shown by Fig. 6, equation-computed $T$ is better than probe-measured $T$ because the former is insensitive to solar radiation. The air moisture measured by an infrared analyzer in CPEC systems must be more accurate than probe-measured air moisture. The better equation-computed $T$ along with better air moisture has no reason not to improve $\rho_d$ estimation from the convention method. The explanations in this paragraph may be expected by Referee #2 to enhance the discussion in section 10.3.1.

*Author revision*
*a.* Add the following to the end of section 10.3.2.
*"Without our exact T equation, in any flux software, either Eqs. (4) or (5) has to be used for sensible heat flux computation. Both equations are approximate ones (see Appendices A and B). Compared to either, our exact equation must be an improvement on the mathematical representation of sensible heat flux. If the equation for sensible heat flux is approximate, then even a perfect measurement gives only an approximate value for flux."*

b. Add one short paragraph to section 10.3.1.
*"Currently, in $CO_2$, $H_2O$ and trace gas flux measurements, mean $\rho_d$ for flux calculations is estimated from T and RH along with P. T and RH are measured mostly by a slow-response T-RH probe without fan-aspiration (e.g., HMP155A. Zhu et al. 2021). As shown in Fig. 6, equation-computed T is better than probe-measured T. The air moisture measured by infrared analyzers in CPEC systems must be more accurate than probe-measured air moisture. The better equation-computed T along with more accurate air moisture has no reason not to improve the estimation for mean $\rho_d$."*

c. Due to citation of Zhu et al. (2021) in b above, add one more reference below line 825.
*"Zhu, J.J., Gao, T., Yu, L.Z., Yu, F.Y., Yang, K., Lu, D.L., Yan, Q.L., Sun, Y.R., Liu, L.F., Xu, S., Zhang, J.X., Zheng, X., Song, L.N., Zhou, X.H. Functions and applications of Multi-tower Platform of Qingyuan Forest Ecosystem Research Station of Chinese Academy of Sciences (Qingyuan Ker Towers). Bulletin of Chinese Academy of Sciences 3, 351-361, 2021."*

3. This study was conducted with only Campbell Scientific instruments. It would be helpful if there was some commentary on use of the proposed technique with other instruments. While Campbell Scientific instruments are widely used for flux measurements using the eddy covariance technique, other companies make 3D sonic anemometers and high frequency gas analyzers that are also widely applied for eddy covariance. Even if brief, any discussion the authors can provide about applicability of the proposed technique with non-Campbell Scientific instruments will make the paper more general. Right now, the information in the paper is specific to only those users who have Campbell Scientific instruments.

*Author response*
For the applications of our developed $T$ equation to any combination of sonic anemometer and infrared analyzer with different models and brands, we developed error equations (24) to (27) to estimate the error of high-frequency $T$ from any combination of sonic and infrared instruments as long as their measurement specifications are given. Our developed $T$ equation (23) including error equation (25) completes this theory, which has not been done from any previous studies (Ishii 1932, Barrett and Suomi 1949, Schotanus et al. 1983, Kaimal and Gaynor 1991, Swiatek 2018).

The $T$ equation [i.e., Eq. (23)] and its error equations [i.e., Eqs. (24) to (27)] that we developed are applicable to any combination of sonic anemometers and infrared analyzers with different models and brands. To implement these improved equations in other instrument combinations will require the hardware specifications of these instruments to complete the necessary modifications to their software.

As this comment indicates, Referee #2 has been aware of the wide applicability of our equations to other CPEC systems. Revision is needed to convey his/her concernt. The applicability is discussed in section 10.2. We propose further clarification in response to this comment in the same way as to Referee #1's comment 8.

*Author revision*
See the Author revision in response to Referee #1's comment 8.

**Editorial Comments**

Beyond these three content recommendations, there are two things that would improve organization of the paper:

1.  Move sections 6 and 7 to an appendix. These sections contain important material, but provide a level of detail that is not essential to the main body of the paper.

*Author response*
This manuscript is a thorough study on high-frequency $T$ derived from $T_s$ and $\chi_{H_2O}$. If the details provided are not enough in main text, readers would feel inconvenient during reading. We moved a large amount of material to the appendices to improve the readability of the main body of the manuscript (Appendices A, B, and C). Because the central goal of the paper was to focus on the high-frequency $T$, we decided not to move sections 6 and 7, since this would frequently send the reader to the appendices and disrupt the reader's brain of thought to learn about this new development.

*Author revision*
Revision is not preferred.

2.  Use headers to better separate the material. For example, section 1 is Introduction, section 2 is Background, and section 10 is Discussion. Following this formant, sections 3 and could be called Theory. Section 5 could be called Materials and Methods. If sections 6 and 7 are not moved to an appendix, they should be included with section 5 under Materials and Methods. Sections 8 and 9 could be called Results.

*Author response*
The suggested headers are concise.

*Author revision*
Adopted suggested headers. The related section numbers were also revised. The section number was reduced from eleven to eight.

3. Some necessary edits.

*Author response*
Many thanks to Referee #2 for his/her, not only quality, but also thorough review. We very much appreciate these comments and improvements.

This version of manuscript was reviewed and revised by our authors, but it did not go through a proofreading process. A proofreading process is needed.

*Author revision*
Brittany Smart professionally proofread this version of manuscript. Her edits were incorporated into the revision. The revision was proofread again by Dr. Xinhua Zhou while checking the consistency throughout for expression

Line 17: temperar should be temperature.

*Author revision*
In line 17, corrected "tempera" into "*temperature*".

Line 20: senosrs should be sensors.
*Author revision*
In line 20, corrected "senosrs" into "*sensors*".

Line 30: CPEC300 is a specific product and needs to be defined (meaning the instruments included with this model should be listed and the manufacturer should be listed).

*Author response*
This line is inside Abstract. The details in model and manufacturer are not preferred in Abstract.

*Author revision*
Revision is not preferred.

Line 44: Panofsky and Dutton (1984) is cited, but is not found in the reference list.
*Author revision*
Between lines 799 and 800, inserted:

Panofsky, H.A., and Dutton, J.A. Atmospheric turbulence: models and method for engineering applications, A Wiley-Interscience Publication, John & Sons, Inc. New York, 397, 1984.

Line 66: suffered to should be changed to suffered by.

*Author revision*
In line 66, corrected "to" to "*by*".

Line 110: contaminated should be contamination.

*Author revision*
In line 110, corrected "contaminated" to "contamination".

Line 152: a should be removed after unmeasurable by.

*Author revision*
In line 110, removed "*a*" after "by".

Lines 243-244: CSAT3A and EC155 are specific Campbell Scientific products, so they should be denoted as such (like at the beginning of the sentence where CPEC310 is denoted as a Campbell Scientific product).

*Author response*
We feel redundant if "(Campbell Scientific Inc., Logan, UT, USA)" is repeated three times in one sentence separately for CPEC300, CSAT3A, and EC155.

*Author revision*
In lines 243, replaced "including" with "*whose major components are*". This revision indicates that CSAT3A and EC155 belong to a CPEC310 system. As long as CPEC310 system is denoted by (Campbell Scientific Inc., Logan, UT, USA), CSAT3A and EC155, as two components of CPEC310, do not need this denotation.

Line 262: Multiple temperature variables are used in equation (27). Subscript c appears to denote calibration, subscript z appears to denote zero, subscript s appears to denote sonic, and unclear what subscript r denotes. Some clarification and definition is required.

*Author response*
Subscript *r* denotes "the range of air temperature". See lines 264 and 265, "….. and $T_{rl}$ and $T_{rh}$ are the low- and the high-end values, respectively, over the operational air temperature range of CPEC systems".  This expression was not clear to readers. Further revision could be better for this expression.

*Author revision*
The above sentence was revised as "……, *subscripts rh and rl indicate the range highest and range lowest values, respectively; and $T_{rh}$ and $T_{rl}$ are the highest- and lowest-T, respectively.*"

Lines 323-324: EC100 is a specific Campbell Scientific product and needs to be denoted as such. It seems the EC100, EC155, and CSAT3A are all components of the CPEC310. If this is the case, it would be helpful if there is a better description of the CPEC310.

*Author response*
We believe that Referee refers to line 325-333. In a CPEC310 system, all components except for barometer are all Campbell Scientific parts. The excepted barometer is labelled with "Freescale Semiconductor, TX, USA". We read through this paragraph several times and feel the description overall is clear after the revision above for line 243.

*Author revision*
See author revision above for line 243.

Line 341: CR6 needs to be defined as a datalogger and denoted as a Campbell Scientific product.

*Author response*
See line 341, "A CR6, supported by EasyFlux-DL-CR6CP (revised version for this study, Campbell Scientific Inc. UT, USA) ………".  It is clear because "CR6" also occurs in EasyFlux "CR6"CP. Additional "Campbell Scientific Inc. UT, USA" in the same sentence is less readable.

Line 357: Sentence needs to be reworded. The phrase even impossible is out of place. Perhaps remove the phrase even impossible from the sentence and then write another sentence to describe how it is impossible to sample fast enough to capture all eddies.

*Author revision*
In line 357, "even impossible" was revised as "*although impossible*".

Line 527: Acronym OPEC is used without being defined. Needs to be defined as open path eddy covariance.

*Author response*
Yes, OPEC in flux community commonly refers "open-path eddy-covariance". In lines 527 and 528, the acronym OPEC is more specifically defined by "(e.g. CSAT3A+EC150 and CSAT3B+LI7500)". This definition may be too specific. The sentence needs rewording.

*Author revision*
In lines 527 and 528, the sentence was revised as "*Some open-path eddy-covariance (OPEC) flux systems (e.g., CSAT3A+EC150 and CSAT3B+LI7500) ……….*".

Line 609: thermometry can be removed.

*Author revision*
In line 609, removed "*-thermometry*".

Line 674: Diving should be dividing by.

*Author revision*
In line 674, corrected "*Diving*" into "*Dividing*".

Lines 682 and 685: expending should be expanding.

*Author revision*
In line 682 and 685, corrected "expending" into "expanding".

**References**

Barrett, E. W. and Suomi, V. E.: Preliminary report on temperature measurement by sonic means, J. Atmos. Sci., 6, 273–276, 1949.

Harrison, R. G., and Burt, S. D.: Quantifying uncertainties in climate data: measurement limitations of naturally ventilated thermometer screens, Environ. Res. Commun., 3, 1-10, doi: 10.1088/2515-7620/ac0d0b, 2021.

Ishii, C.: Supersonic velocity in gases: especially in dry and humid air. Scientific Papers of the Institute of Physical and Chemical Research, Tokyo, 26, 201–207 pp., 1932.

Kaimal, J. C., and Gaynor, J. E.: Another look to sonic thermometry, Boundary-Layer Meteorol., 56, 401-410, 1991.

Mauder, M., and Zeeman, M. J.: Field intercomparison of prevailing sonic anemometers, Atmos. Meas. Tech., 11, 249-263, doi: 10.5194/amt-11-249-2018, 2018.

Schotanus, P., Nieuwstadt, F. T. M., and DeBruin, H. A. R.: Temperature measurement with a sonic anemometer and its application to heat and moisture fluctuations, Boundary-Layer Meteorol., 26, 81-93, 1983.

Swiatek, E: Derivation of Temperature (Tc) from the Sonic Virtual Temperature (Ts), vapor density ($\rho$v)/vapor pressure (e) and pressure (P). Campbell Scientific Inc. Logan, UT, 1-5 pp., 2018

WMO: Guide to Instruments and Methods of Observation, WMO-No. 8, Volume I - Measurement of Meteorological Variables, World Meteorological Organization, Geneva, 548 pp., 2018.

van Dijk, A.: The principles of surface flux physics. Department of Meteorology and Air Quality, Agriculture University Wageningen, 40–41 pp., 2002.

Zhou, X., Yang, Q., Zhen, X., Li, Y., Hao, G., Shen, H., Gao, T., Sun, Y., and Zheng, N.: Recovery of the three-dimensional wind and sonic temperature data from a physically deformed sonic anemometer, Atmos. Meas. Tech., 11, 5981–6002, doi:10.5194/amt-11-5981-2018, 2018.

Zhu, J.J., Gao, T., Yu, L.Z., Yu, F.Y., Yang, K., Lu, D.L., Yan, Q.L., Sun, Y.R., Liu, L.F., Xu, S., Zhang, J.X., Zheng, X., Song, L.N., Zhou, X.H. Functions and applications of Multi-tower Platform of Qingyuan Forest Ecosystem Research Station of Chinese Academy of Sciences (Qingyuan Ker Towers). Bulletin of Chinese Academy of Sciences 3, 351-361, 2021.

---

## Author Response (AR2)

**INSTITUTE OF APPLIED ECOLOGY, CHINESE ACADEMY OF SCIENCES**

**72 Wenhua Road, Shenyang, Liaoning, 110016, China**

Nov 12, 2021

RE: Adjustment and corrections for amt-2021-160R

Dr. Keding Lu
College of Environmental Science and Engineering
Peking University
Beijing 100871, China

Dear Dr. Lu,

Thank you so much for your favorable decision regarding our manuscript, "Air temperature equation derived from sonic temperature and water vapor mixing ratio for turbulent air flow through closed-path eddy-covariance flux systems," for publication in *Atmospheric Measurement Techniques* (AMT).  Taking your comments into account, we incorporated our adjustments and corrections into the manuscript. Afterward, Ms. Brittney Smart professionally proofread the full manuscript again. After her proofreading, Drs. Takle and Zhou conducted a final read-through, checking throughout for consistent expressions with AMT requirements.

Our adjustments and corrections in response to your comments are addressed below.

We appreciate your consideration in the publication of our manuscript in AMT.

Sincerely,

Tian Gao, Ph.D., Research Associate Professor
Remote Sensing for Forest Fluxes and Management

**Adjustments and corrections in response to Associate Editor on "Air temperature equation derived from sonic temperature and water vapor mixing ratio for turbulent air flow sampled through closed-path eddy-covariance flux systems"**

X.H. Zhou, T. Gao, E.S. Takle, X.J. Zhen, A.E Suyker, T. Awada, J. Okalebo, J.J. Zhu

**Associate Editor's comment**

All the references cited in the comments of Referee #1 are worth comment and citation in the revised paper.

**Response**

Referee #1 cited five references: Schotanus et al. (1983), Kaimal and Gaynor (1991), Harrison and Burt (2021), Mauder and Zeeman (2018), and WMO (2018).

The first two are closely related to our study topic. Both were deeply discussed in sections 1 and 2 and appendices A and B in all versions of this manuscript. Although results from the other three references were used to inform the text, these three were not explicitly cited. Following your comments, the conclusions from the three references are further discussed and are now explicitly cited in two paragraphs of the manuscript.

**Author adjustments and corrections**
*(Line numbers used below refer to those in version: _amt_2021_160R)*

1. Harrison and Burt (2021) and WMO (2018).
   The paragraph between lines 105 and 111 is revised as:

   Measurements of $T$ at high frequency (similar to those at low frequency) are contaminated by solar radiation, even under shields (Lin et al., 2001) and when aspirated (Campbell Scientific Inc., 2010; R.M. Young Company, 2004; Apogee Instruments Inc., 2013; Blonquist and Bugbee, 2018). Although a naturally ventilated or fan-aspirated radiation shield could ensure the accuracy of a conventional (i.e., slow-response) thermometer often within ±0.2 K at 0 ℃ (Harrison and Burt, 2021) to satisfy the standard for conventional $T$ measurement as required by the World Meteorological Organization (WMO, 2018), the aspiration shield method cannot acquire $T$ at high frequency due to the disturbance of an aspiration fan and the blockage of a shield to natural turbulent flows. Additionally, fine wires have limited applicability for long-term measurements in rugged field conditions typically encountered in ecosystem monitoring.

2. Mauder and Zeeman (2018).
   The three sentences between lines 242 and 248 are revised as:

   ……… Sonic anemometers and infrared analyzers with different models and brands have different specifications from their manufacturers. The manufacturer of the anemometer we studied employs carbon fiber with minimized thermo-expansion and -contraction for sonic strut stability (via personal communication with CSAT structural designer Antoine Rousseau, 2021); structural design with optimized sonic volume for less aerodynamic disturbance (Fig. 1); and advanced proprietary sonic firmware for more accurate measurements (Zhou et al. 2018), which reduces the variability of $T_s$ by several Kelvin compared to what has been reported for sonics from other models (Mauder and Zeeman, 2018). Any combination of sonic and infrared instruments has a combination of the $\Delta T_s$ and $\Delta \chi_{H2O}$, which are specified by their manufacturers. In turn, from Eq. (25), the combination generates $\Delta T$ of equation-computed $T$ for the corresponding combination of the sonic and infrared instruments with given models and brands. Therefore, Eqs. (23) and (25) are applicable to any CPEC system beyond our study brand. The applicability of Eq. (23) for any sonic or infrared instrument can be assessed based on $\Delta T$ against the required $T$ accuracy for a specific application.

3. Insert the three references into the References section.

    a.   Between Lines 799 and 800 is inserted:

    Harrison, R.G. and Burt, S.D.: Quantifying uncertainties in climate data: measurement limitations of naturally ventilated thermometer screens, Environ. Res. Commun., 3, 1–10, https//doi.or/10.1088/2515-7620/ac0d0b, 2021.

    b.   Between Lines 837 and 838 is inserted:

    Mauder, M. and Zeeman, M.J.: Field intercomparison of prevailing sonic anemometers, Atmos. Meas. Tech., 11, 249–263, https//doi.or/10.5194/amt-11-249-2018, 2018.

    c.   Between Lines 871 and 872 is inserted:

    WMO: Guide to Instruments and Methods of Observation, WMO-No. 8, Volume I — Measurement of Meteorological Variables, World Meteorological Organization, Geneva, 548 p., 2018.

4. Additional proofreading.

This manuscript was once more proofread and checked in its entirety. Some minor corrections are indicated in the latest submitted version with change trackers.

**References**

Harrison, R.G. and Burt, S.D.: Quantifying uncertainties in climate data: measurement limitations of naturally ventilated thermometer screens, Environ. Res. Commun., 3, 1–10, doi: 10.1088/2515-7620/ac0d0b, 2021.

Kaimal, J.C. and Gaynor, J.E.: Another look at sonic thermometry, Boundary-Layer Meteorol., 56, 401–410, 1991.

Mauder, M. and Zeeman, M.J.: Field intercomparison of prevailing sonic anemometers, Atmos. Meas. Tech., 11, 249–263, doi: 10.5194/amt-11-249-2018, 2018.

Schotanus, P., Nieuwstadt, F.T.M., and de Bruin, H.A.R.: Temperature measurement with a sonic anemometer and its application to heat and moisture fluctuations, Boundary-Layer Meteorol., 26, 81–93, 1983.

WMO: Guide to Instruments and Methods of Observation, WMO-No. 8, Volume I — Measurement of Meteorological Variables, World Meteorological Organization, Geneva, 548 p., 2018.